# Teaching Molecular Dynamics to a Non-Autoregressive Ionic Transport Predictor

Jiyeon Kim [1]  Byungju Lee [2 3]  Won-Yong Shin [1]

## Abstract

Unlike most static material properties widely studied in the machine learning literature, ionic transport properties are inherently dynamic, making their fast and accurate prediction from static atomic structures challenging. The current standard approach, molecular dynamics (MD) simulations, suffers from prohibitively high computational cost. Recent autoregressive learning-based MD acceleration methods requiring sequential inference remain slow and prone to error accumulation; in contrast, existing non-autoregressive material property prediction models are less accurate because they fail to exploit dynamics. Moreover, existing methods typically benefit from datasets either with or without atomic trajectories, but not both. To overcome these limitations, we propose a non-autoregressive learning framework based on auxiliary modality learning, which treats atomic *trajectories* as an auxiliary modality during training but does not require them at inference. This enables the predictor to learn dynamics without sequential inference while benefiting from both types of datasets. As a result, our framework achieves over $200\times$ speedup compared to autoregressive models on the dataset with atomic trajectories while substantially reducing prediction error relative to non-autoregressive benchmarks across both types of datasets. Our code is available at https://github.com/jykim-git/MD.git.

## 1. Introduction

**Dynamics from statics.** Predicting ionic transport properties from equilibrium atomic structures is fundamentally

---

[1]School of Mathematics and Computing (Computational Science and Engineering), Yonsei University, Seoul, Republic of Korea [2]Korea Institute of Science and Technology (KIST), Seoul, Republic of Korea [3]Nanoscience and Technology, KIST School, University of Science and Technology, Seoul, Republic of Korea. Correspondence to: Won-Yong Shin <wy.shin@yonsei.ac.kr>.

*Proceedings of the 43rd International Conference on Machine Learning*, Seoul, South Korea. PMLR 306, 2026. Copyright 2026 by the author(s).

challenging because it requires inferring long-time atomic dynamics (Gao et al., 2025; Shewmon, 2016) from static input lacking temporal information, thereby creating a severe modality gap between the input and the target. Despite its central importance in materials science, especially for rechargeable batteries (Orikasa et al., 2016; Itou et al., 2025), this task remains underexplored in the machine learning (ML) literature. Most benchmark datasets (Dunn et al., 2020; Choudhary et al., 2020; Ruddigkeit et al., 2012; Ramakrishnan et al., 2014) for material prediction focus on static properties obtained from quantum mechanical simulations of single atomic configurations. In contrast, ionic transport properties are conventionally computed by molecular dynamics (MD) simulations (Gao et al., 2020; Xu et al., 2023; Schuett et al., 2025; Shi et al., 2025), which generate atomic trajectories from equilibrium structures. However, the high computational cost of MD severely limits large-scale materials screening for ionic transport properties. Although there exists an MD benchmark, such as MD17 (Chmiela et al., 2017), its targets are instantaneous quantities—energies and forces determined by an atomic configuration at a certain time step—whereas ionic transport depends on long-time atomic motion.

**Cost and accuracy.** This modality gap makes it difficult for both autoregressive and non-autoregressive approaches to simultaneously achieve low computational cost and high accuracy. An autoregressive MD acceleration method (Nam et al., 2025; Schreiner et al., 2023) sequentially generates atomic trajectories using generative models, leading to slow inference and error accumulation that can cause trajectory divergence. Conversely, one may attempt to apply conventional non-autoregressive material property prediction models (Xie & Grossman, 2018; Schütt et al., 2017; Chen et al., 2019; Du et al., 2024; Yan et al., 2022; 2024) to map equilibrium structures directly to ionic transport properties in a single forward pass, enabling fast inference. Nevertheless, a straightforward application yields limited accuracy because atomic trajectories that encode dynamical information cannot be used as input.

**Data scarcity.** Neither approach can fully exploit limited ionic transport datasets, which are scarce due to expensive experiments or long MD simulations. Ionic transport datasets often entirely lack atomic trajectories due to their substantial storage requirements or the impracticality of

obtaining atomic trajectories experimentally. However, autoregressive approaches (Nam et al., 2025; Schreiner et al., 2023) typically require atomic trajectories and are therefore inapplicable to structure-based datasets that lack such information. In contrast, existing non-autoregressive material property prediction models (Xie & Grossman, 2018; Schütt et al., 2017; Chen et al., 2019; Du et al., 2024; Yan et al., 2022; 2024) fail to fully exploit dynamical information even when a trajectory-based dataset is available.

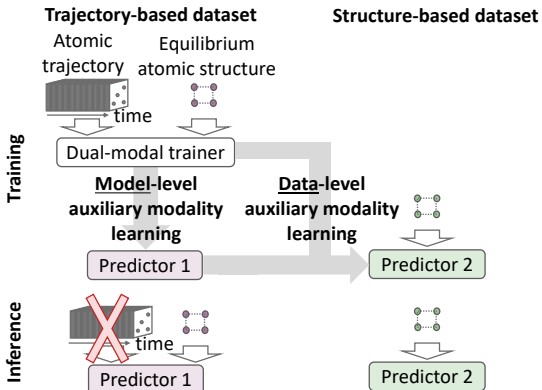

*Figure 1.* Overview of our non-autoregressive learning framework with auxiliary modality learning. A dual-modal trainer leverages both atomic trajectories and equilibrium structures during training, while the predictors for trajectory-based and structure-based datasets operate only on equilibrium structures at inference. Model-level auxiliary modality learning transfers dynamical knowledge from the dual-modal trainer to Predictor 1 via closed-form initialization, enabling accurate trajectory-free prediction. Data-level auxiliary modality learning further transfers knowledge to structure-based datasets by initializing Predictor 2 from both the dual-modal trainer and Predictor 1.

**Our non-autoregressive learning framework.** We address these challenges by introducing a non-autoregressive learning framework based on auxiliary modality learning (Shen et al., 2023). We treat equilibrium atomic structures and atomic trajectories as two distinct input modalities and use atomic trajectories as auxiliary information during training while removing them at inference. As described in Figure 1, our framework consists of two components: **model-level auxiliary modality learning** and (optionally) **data-level auxiliary modality learning**. First, model-level auxiliary modality learning transfers dynamical knowledge from a dual-modal trainer to the non-autoregressive Predictor 1 by enforcing Predictor 1 to reproduce the hidden representations of the dual-modal trainer without atomic trajectories. In this way, Predictor 1 can make accurate predictions without atomic trajectories as input. Second, when training on structure-based datasets is desired, we introduce data-level auxiliary modality learning as an optional component for structure-based datasets where atomic trajectories are unavailable; it transfers knowledge from a trajectory-based dataset by initializing Predictor 2 from both the dual-modal trainer and Predictor 1. This enables models trained on

structure-based datasets to benefit from information learned from the trajectory-based dataset.

**Key aspects of our auxiliary modality learning.** Although auxiliary modality learning (Shen et al., 2023) has been previously studied, to the best of our knowledge, this is the first work to formulate atomic trajectories as a privileged modality for ionic transport prediction in materials science. From a technical perspective, we introduce a closed-form initialization based on ridge regression that enables Predictor 1 to reproduce the hidden representations of the dual-modal trainer in a data-efficient manner. Compared to conventional distillation approaches (Hinton et al., 2015) based on iterative gradient optimization, the proposed initialization is particularly effective under data-scarce settings such as ionic transport prediction. Furthermore, we leverage scientific foundation models (Park et al., 2024; Goswami et al., 2024), originally developed for interatomic force prediction and time-series analysis, to extract informative embeddings from equilibrium atomic structures and atomic trajectories. By doing so, our framework exploits rich physical priors encoded in the pretrained foundation models without requiring task-specific fine-tuning, enabling accurate prediction even with limited training data.

**Highlights of results and analyses.** We comprehensively evaluate the performance on multiple datasets. On a trajectory-based dataset, our framework achieves up to $200\times$ faster inference than the state-of-the-art autoregressive approach (Nam et al., 2025), while substantially improving prediction accuracy. We also observe a significant reduction in mean absolute error (MAE) on $\log_{10}$-scaled targets compared to non-autoregressive benchmarks (Yan et al., 2022; 2024; Du et al., 2024). Notably, we consistently observe large error reductions on structure-based datasets, including a real-world experimental dataset. Our main contributions are as follows:

• **Dynamics-aware non-autoregressive prediction.** We present a non-autoregressive learning framework that employs auxiliary modality learning to capture atomic dynamics during training while enabling trajectory-free inference.

• **Full exploitation of limited datasets.** We introduce a closed-form initialization, which enables effective knowledge transfer for model-level auxiliary modality learning in data-scarce regimes. Furthermore, our framework optionally includes data-level auxiliary modality learning for effective transfer of dynamical knowledge from a trajectory-based dataset to structure-based datasets, even when the datasets differ in target properties, data sources, or ionic species.

• **Improved accuracy and inference speed.** Our approach achieves substantial improvements in both prediction accuracy and inference speed over existing autoregressive and non-autoregressive benchmarks across various datasets.

## 2. Background

**Ionic transport properties.** Ionic transport plays a crucial role in rechargeable batteries, where ions shuttle between electrodes during charging and discharging (Goodenough, 2018). This ionic motion directly influences key performance metrics, including charging speed, power density, energy efficiency, and lifetime (Logan & Dahn, 2020; Itou et al., 2025). Despite their importance, ionic transport properties remain relatively underexplored in the ML literature.

**MD simulation.** This fundamental dependence on atomic dynamics makes MD simulations the most dominant approach for estimating ionic transport by explicitly simulating atomic motion (Gao et al., 2020; Xu et al., 2023; Schuett et al., 2025; Shi et al., 2025). MD simulations iteratively update atomic positions by computing interatomic forces and numerically integrating equations of motion, producing detailed atomic trajectories from which ionic transport properties can be extracted (Tuckerman & Martyna, 2000). Despite their accuracy, MD simulations are computationally expensive. They require extremely small integration time steps (*e.g.*, $\sim 10^{-15}$ s) to maintain numerical stability (Zheng et al., 2021). Moreover, in solid materials, ions spend most of their time oscillating around equilibrium positions, while transport-relevant diffusion events occur only rarely (Gao et al., 2025; Shewmon, 2016). Capturing such rare events therefore demands very long simulations. Although machine-learned interatomic potentials (MLIPs) (Merchant et al., 2023; Lee et al., 2025; Park et al., 2024) significantly reduce the cost of force evaluations, even MLIP-based MD simulations can still require several hours to predict ionic transport properties for a single material (Nam et al., 2025). As alternatives to MD, autoregressive MD acceleration methods and non-autoregressive material property prediction methods are reviewed in Appendix A.

**Auxiliary modality learning.** In learning using privileged information (LUPI) (Vapnik & Vashist, 2009), a teacher leverages additional information available only during training and not at inference time. The LUPI framework was later unified with knowledge distillation (Hinton et al., 2015) under the concept of generalized distillation (Lopez-Paz et al., 2015), and further extended to auxiliary modality learning (Shen et al., 2023), where extra modalities are exploited during training but discarded at inference.

## 3. Methodology[1]

We begin by outlining the key features of our framework.

---

[1]The notations in this section, the pseudocodes, and the implementation details for our learning framework with auxiliary modality learning are available in Appendices B, C, and E, respectively.

**Non-autoregressive prediction (Section 3.1).** Rather than sequentially (autoregressively) predicting atomic trajectories (time series of atomic positions) and subsequently deriving ionic transport properties from them (Nam et al., 2025), we formulate the task as non-autoregressively predicting an ionic transport property of a material from its equilibrium atomic structure and temperature. This formulation avoids error accumulation and significantly reduces inference time.

**Model-level auxiliary modality learning (Section 3.4).** A central challenge of this formulation is how to effectively exploit rich dynamic information in atomic trajectories when the predictor is non-autoregressive and does not take trajectories as input. Utilizing atomic trajectories at inference time is prohibitive due to their high generation cost. Our key idea is to treat atomic trajectories as an additional modality that is available only during training. As illustrated in Figure 1, model-level auxiliary modality learning aims to transfer knowledge from a dual-modal trainer that utilizes both the atomic trajectories and equilibrium atomic structures to a predictor that operates without the atomic trajectories. To this end, we initialize the predictor with a closed-form linear regression solution that aligns the hidden representations of the dual-modal trainer and the predictor, enabling accurate and fast knowledge transfer without iterative gradient-based optimization when atomic trajectories are scarce.

**(Optional) Data-level auxiliary modality learning (Section 3.5).** We further introduce data-level auxiliary modality learning, an optional component designed to handle the practical scenario in which many ionic transport datasets entirely lack atomic trajectories. Because the predictor's encoder is constrained by the closed-form initialization, its hidden representations may not generalize to structure-based datasets. Instead, as depicted in Figure 1, the encoder for structure-based datasets is initialized with the dual-modal trainer's structure encoder. Meanwhile, the decoder is initialized with the predictor's decoder, enabling knowledge learned from a trajectory-based dataset to assist learning on a structure-based dataset.

### 3.1. Problem Definition

**Formulation.** We aim to learn a mapping

$$f(\mathbf{x}, T) \;\mapsto\; y_s, \tag{1}$$

where $\mathbf{x} \in \mathbb{R}^{N \times 3}$ denotes the equilibrium atomic structure, represented by the 3-dimensional (3D) coordinates of $N$ atoms corresponding to a stable (energy-minimized) configuration, $T$ denotes the temperature, and $f$ represents a predictor that non-autoregressively predicts an ionic transport property $y_s$ of ionic species $s$. Notably, atomic trajectories are not included as input to the predictor $f$.

**Ionic transport property $y_s$.** The ionic transport property $y_s$ characterizes the mobility of a given ionic species $s$ in

the material at temperature $T$. Depending on the datasets, $y_s$ may correspond to the mean squared displacement at the final simulation time $MSD_s$ (Nam et al., 2025), the ionic diffusivity $D_s$ (Zheng et al., 2024), or the ionic conductivity $\sigma_s$ (Therrien et al., 2025). These quantities provide different but closely related descriptions of the same underlying ionic transport process and are therefore transferable within a unified learning framework. The explicit definitions of these quantities and their physical relationships are provided in Appendix D. By formulating these related transport quantities within a unified learning framework, our approach allows knowledge to be shared across datasets with different target definitions, fully exploiting scarce ionic transport datasets.

### 3.2. Dataset Types

Datasets are categorized into two types based on the availability of atomic trajectories $\mathbf{p}$, as summarized in Table 1.

*Table 1.* Summary of the two types of datasets. Here, O and X indicate that the corresponding information is available and unavailable, respectively.

| DATASET TYPE | | $\mathbf{p}$ | $\mathbf{x}$ | $T$ | $y_s$ |
|---|---|---|---|---|---|
| $\mathcal{D}^{trj}$ | TRAIN | O | O | O | O |
| $\mathcal{D}^{trj}$ | TEST | X | O | O | O |
| $\mathcal{D}^{str}$ | TRAIN | X | O | O | O |
| $\mathcal{D}^{str}$ | TEST | X | O | O | O |

**Trajectory-based dataset $\mathcal{D}^{trj}$.** Each sample in a trajectory-based dataset $\mathcal{D}^{trj}$ consists of an atomic trajectory $\mathbf{p} \in \mathbb{R}^{L \times N \times 3}$, which is a sequence of 3D coordinates of $N$ atoms over $L$ time steps, along with the corresponding equilibrium atomic structure $\mathbf{x}$, temperature $T$, and ionic transport property $y_s$. Notably, the atomic trajectory $\mathbf{p}$ is available only during training and is not used at test time.

**Structure-based dataset $\mathcal{D}^{str}$.** However, ionic transport properties are often reported without atomic trajectories and it is impractical to obtain atomic trajectories $\mathbf{p}$ experimentally. In other words, some ionic transport datasets lack atomic trajectories $\mathbf{p}$ and contain only equilibrium atomic structures $\mathbf{x}$, temperatures $T$, and ionic transport properties $y_s$. Such datasets are referred to as structure-based datasets $\mathcal{D}^{str}$ in our study.

### 3.3. Input Embeddings

**Trajectory embedding $\mathbf{E}_{\mathbf{p}}^i$.** Based on the observation that ions in solid materials predominantly vibrate around equilibrium positions and occasionally undergo hopping events that drive long-range transport (Rodin et al., 2022; Gustafsson et al., 2024), we transform atomic trajectories of target ionic species into the frequency domain using a Fourier transform, which helps distinguish hopping dynamics from vibrational motions. To obtain a compact yet informative representa-

tion under data scarcity, we leverage MOMENT (Goswami et al., 2024), a foundation model for time-series analysis, to condense the trajectory spectrum into a fixed-dimensional trajectory embedding $\mathbf{E}_{\mathbf{p}}^i$ for a data sample index $i$.

**Structure embedding $\mathbf{E}_{\mathbf{x}}^i$.** To extract informative structural representations under limited data, we employ SevenNet (Park et al., 2024), an MLIP foundation model originally developed for force calculation in MD. SevenNet produces node embeddings for each atom and edge embeddings for each interatomic interaction. Following a strategy similar to MeshGraphNet (Pfaff et al., 2020), we aggregate local structural information by concatenating node and edge embeddings and averaging over neighboring atoms to obtain an atomic embedding $\mathbf{E}_a^i$:

$$(\mathbf{E}_a^i)_{k,:} = \frac{1}{|\mathcal{N}^i(k)|} \sum_{l \in \mathcal{N}^i(k)} \left[ \mathbf{n}_k^i; \mathbf{m}_{kl}^i; \mathbf{n}_l^i \right], \quad (2)$$

where $(\mathbf{E}_a^i)_{k,:}$ denotes atomic embedding corresponding to the $k$-th atom of the $i$-th data sample; $\mathcal{N}^i(k)$ denotes the set of neighboring atoms of the $k$-th atom of the $i$-th data sample; $\mathbf{n}_k^i$ and $\mathbf{n}_l^i$ denote the node embeddings of the $k$-th and $l$-th atoms of the $i$-th data sample, respectively; and $\mathbf{m}_{kl}^i$ denotes the edge embedding corresponding to the interaction between the $k$-th and $l$-th atoms of the $i$-th data sample. Then, we select only the rows of the atom-wise embedding $\mathbf{E}_a^i$ corresponding to target ion species $s$ to obtain $\mathbf{E}_{a,s}^i$.

Under extreme data scarcity, we found that a lightweight encoder with strong pretrained embeddings is more robust than training a deeper encoder. To minimize overfitting due to limited training data and to facilitate fast inference, we intentionally adopt linear encoders to extract hidden representations, to be specified in Section 3.4. To compensate for the limited expressiveness of a purely linear mapping, we introduce nonlinearity at the embedding level through polynomial expansion:

$$\mathbf{E}_{\mathbf{x}}^i = \left[ \mathbf{E}_{a,s}^i; \; \mathbf{E}_{a,s}^i \odot \mathbf{E}_{a,s}^i; \; \mathbf{E}_{a,s}^i \odot \mathbf{E}_{a,s}^i \odot \mathbf{E}_{a,s}^i \right], \quad (3)$$

where $\odot$ denotes element-wise multiplication and ; denotes concatenation. We use a third-order expansion to balance expressive capacity and embedding dimensionality.

**Temperature embedding $\mathbf{E}_T^i$.** To model nonlinear temperature effects while remaining compatible with the linear encoder without a bias term, we again apply the polynomial expansion to the normalized temperature:

$$\mathbf{E}_T^i = \left[ 1; \; \frac{T^i}{T_m}; \; \left(\frac{T^i}{T_m}\right)^2; \; \left(\frac{T^i}{T_m}\right)^3 \right], \quad (4)$$

where $T^i$ is the temperature of the $i$-th data sample, $T_m$ is a normalization constant. The first term enables the linear encoder to capture a constant offset in the absence of bias.

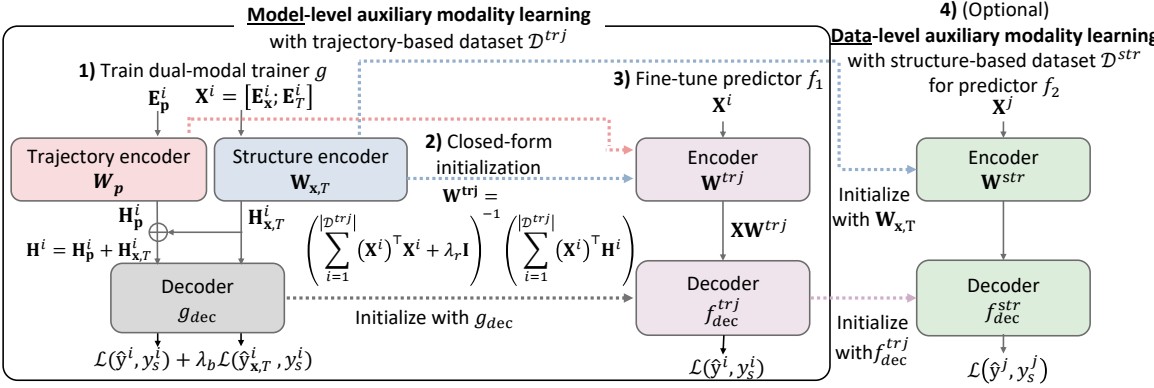

*Figure 2.* Training procedure of the proposed framework with auxiliary modality learning. **1)** A dual-modal trainer $g$ is trained on a trajectory-based dataset using trajectory, structure, and temperature embeddings $\mathbf{E}_{\mathbf{p}}^i$, $\mathbf{E}_{\mathbf{x}}^i$, and $\mathbf{E}_T^i$. **2)** Trajectory-informed knowledge is transferred from the dual-modal trainer $g$ to a predictor $f_1$ via closed-form initialization. **3)** The predictor $f_1$ is fine-tuned on the trajectory-based dataset $\mathcal{D}^{trj}$ without trajectory inputs. **4)** Optionally, for transfer to structure-based datasets $\mathcal{D}^{str}$, data-level auxiliary modality learning initializes a new encoder $\mathbf{W}^{str}$ from the structure encoder $\mathbf{W}_{\mathbf{x},T}$ of $g$ and reuses the decoder $f_{dec}^{trj}$; then, the predictor $f_2$ is trained on the structure-based dataset $\mathcal{D}^{str}$.

## 3.4. Model-Level Auxiliary Modality Learning

To leverage the rich dynamical information contained in the trajectory embedding $\mathbf{E}_{\mathbf{p}}^i$ during training while enabling trajectory-free inference, we introduce a model-level auxiliary modality learning scheme. As illustrated in Figure 2, we first train a dual-modal trainer $g$ on a trajectory-based dataset $\mathcal{D}^{trj}$ using three embeddings, $\mathbf{E}_{\mathbf{p}}^i$, $\mathbf{E}_{\mathbf{x}}^i$, and $\mathbf{E}_T^i$. We then transfer its knowledge to a predictor $f_1$ via closed-form initialization. The predictor $f_1$ is subsequently fine-tuned on the trajectory-based dataset $\mathcal{D}^{trj}$ without trajectory embeddings $\mathbf{E}_{\mathbf{p}}^i$.

**Dual-modal trainer $g$.** The dual-modal trainer $g$ consists of a trajectory encoder $\mathbf{W}_{\mathbf{p}}$ and a structure encoder $\mathbf{W}_{\mathbf{x},T}$, both implemented as single linear layers. The trajectory encoder $\mathbf{W}_{\mathbf{p}}$ maps a trajectory embedding $\mathbf{E}_{\mathbf{p}}^i$ to a trajectory-based hidden representation $\mathbf{H}_{\mathbf{p}}^i$, while the structure encoder $\mathbf{W}_{\mathbf{x},T}$ maps a structure–temperature embedding $\mathbf{X}^i = [\mathbf{E}_{\mathbf{x}}^i; \mathbf{E}_T^i]$ to a structure-based hidden representation $\mathbf{H}_{\mathbf{x},T}^i$:

$$\begin{aligned} \mathbf{H}_{\mathbf{p}}^i &= \mathbf{E}_{\mathbf{p}}^i \mathbf{W}_{\mathbf{p}}, \\ \mathbf{H}_{\mathbf{x},T}^i &= \mathbf{X}^i \mathbf{W}_{\mathbf{x},T}. \end{aligned} \quad (5)$$

The two hidden representations are combined additively to obtain the hidden representation $\mathbf{H}^i$,

$$\mathbf{H}^i = \mathbf{H}_{\mathbf{p}}^i + \mathbf{H}_{\mathbf{x},T}^i, \quad (6)$$

and are passed to a multi-layer perceptron (MLP)-based decoder $g_{dec}$ to predict $\hat{y}^i$.

**Regularized training of dual-modal trainer $g$.** The training loss for the dual-modal trainer $g$ includes $L_1$ loss between its prediction $\hat{y}^i$ and the ground truth $y_s^i$. However, atomic trajectories typically provide stronger supervision than equilibrium atomic structures. As a result, the dual-modal trainer $g$ may overly rely on the trajectory encoder

$\mathbf{W}_{\mathbf{p}}$. To ensure that the structure encoder learns informative representations during training, we introduce an additional regularization term when training the dual-modal trainer $g$. Specifically, we require the dual-modal trainer $g$ to produce accurate predictions using only the structure-based hidden representation $\mathbf{H}_{\mathbf{x},T}^i$. This constraint prevents the model from overly relying on the trajectory encoder $\mathbf{W}_{\mathbf{p}}$. More concretely, we generate an auxiliary prediction $\hat{y}_{\mathbf{x},T}^i$ from the structure-based hidden representation $\mathbf{H}_{\mathbf{x},T}^i$ using the same decoder $g_{dec}$ and add the corresponding loss to the original loss based on $\hat{y}^i$:

$$\mathcal{L}(\hat{y}^i, y_s^i) + \lambda_b \mathcal{L}(\hat{y}_{\mathbf{x},T}^i, y_s^i), \quad (7)$$

where $\mathcal{L}$ denotes the $L_1$ loss and $\lambda_b$ is a balancing parameter.

**Closed-form initialization of predictor $f_1$.** Our goal is to enforce the predictor $f_1$ to reproduce the combined hidden representation $\mathbf{H}^i$ learned by the dual-modal trainer $g$, even in the absence of the trajectory embedding $\mathbf{E}_{\mathbf{p}}^i$, under limited trajectory data. To this end, as depicted in Figure 2, we initialize the encoder $\mathbf{W}^{trj}$ of the predictor $f_1$ by solving the following ridge regression problem (Hoerl & Kennard, 1970):

$$\min_{\mathbf{W}^{trj}} \sum_{i=1}^{|\mathcal{D}^{trj}|} \left\| \mathbf{X}^i \mathbf{W}^{trj} - \mathbf{H}^i \right\|_F^2 + \lambda_r \left\| \mathbf{W}^{trj} \right\|_F^2, \quad (8)$$

where $\| \cdot \|_F$ denotes the Frobenius norm, $\lambda_r \in \mathbb{R}_{>0}$ is a ridge regularization parameter that controls the trade-off between faithfully matching the hidden representations and preventing overfitting under scarce trajectory data, and $|\mathcal{D}^{trj}|$ denotes the total number of data samples in the trajectory-based dataset $\mathcal{D}^{trj}$.

This optimization results in the following closed-form solu-

*Table 2.* Summary of datasets.

| DATASET | TYPE | GENERATION | SPECIES | TARGET | TEMPERATURES (K) |
|---|---|---|---|---|---|
| 1 (NAM ET AL., 2025) | $\mathcal{D}^{trj}$ | MD | Li | $\text{MSD}_{\text{Li}}$ | 600, 800, 1000, 1200 |
| 2 (ZHENG ET AL., 2024) | $\mathcal{D}^{str}$ | MD | 61 SPECIES (TRAIN) Na (TEST) | $\text{D}_s$ (TRAIN) $\text{D}_{\text{Na}}$ (TEST) | 1000, 1500, 2000, 2500 |
| 3 (THERRIEN ET AL., 2025) | $\mathcal{D}^{str}$ | EXPERIMENTS | Li | $\sigma_{\text{Li}}$ | 300 |

tion (Hoerl & Kennard, 1970):

$$\mathbf{W}^{trj} = \left( \sum_{i=1}^{|\mathcal{D}^{trj}|} (\mathbf{X}^i)^{\top} \mathbf{X}^i + \lambda_r \mathbf{I} \right)^{-1} \left( \sum_{i=1}^{|\mathcal{D}^{trj}|} (\mathbf{X}^i)^{\top} \mathbf{H}^i \right),$$

(9)

where $\mathbf{I}$ is the identity matrix. This closed-form initialization employs direct solvers (*e.g.,* Cholesky factorization (Benoît, 1924)), avoiding gradient-based iterative optimization. Direct solvers compute the solution to the resulting linear system up to floating-point precision (Higham, 2002; Golub & Van Loan, 1996), yielding accurate hidden-representation matching. While iterative optimization can be computationally cheaper than direct solvers when the number of training samples is large, ionic transport datasets are typically small. In this data-scarce regime, the direct solver is computationally affordable and provides a more accurate solution than iterative optimization. In addition, the decoder $f_{\text{dec}}^{\text{trj}}$ of the predictor $f_1$ is initialized using the decoder $g_{\text{dec}}$ of the dual-modal trainer $g$, which has been trained with access to atomic trajectory information.

**Fine-tuning of predictor $f_1$.** After separately initializing the encoder $\mathbf{W}^{\text{trj}}$ and the decoder $f_{\text{dec}}^{\text{trj}}$, we fine-tune them on the trajectory-based dataset while omitting the trajectory input $\mathbf{p}$, thereby adapting the model to the trajectory-based dataset as a whole. As illustrated in Figure 2, the model receives structure–temperature embeddings $\mathbf{X}^i$ and is optimized by minimizing $L_1$ loss between its prediction $\hat{y}^i$ and the ground truth $y_s^i$. This step allows the predictor $f_1$ to refine its parameters while preserving the trajectory-information injected through the closed-form initialization.

### 3.5. Data-Level Auxiliary Modality Learning

While model-level auxiliary modality learning alone is sufficient to fully exploit the trajectory-based dataset $\mathcal{D}^{trj}$, it cannot be directly applied to a structure-based dataset $\mathcal{D}^{str}$. When training on the structure-based dataset $\mathcal{D}^{str}$ is desired, data-level auxiliary modality learning can be optionally used to transfer knowledge learned from the trajectory-based dataset $\mathcal{D}^{trj}$.

**Initialization of predictor $f_2$.** A key challenge is that the encoder $\mathbf{W}^{trj}$ of the predictor $f_1$ often becomes biased toward the trajectory-based data distribution during closed-form initialization, as it is forced to reproduce the hidden representations learned by the dual-modal trainer $g$. Directly reusing this encoder $\mathbf{W}^{trj}$ can therefore hinder generalization to structure-based datasets. To address this

issue, we initialize a new encoder $\mathbf{W}^{str}$ using the structure encoder $\mathbf{W}_{\mathbf{x},T}$ of the dual-modal trainer $g$, as depicted in Figure 2. The structure encoder $\mathbf{W}_{\mathbf{x},T}$ is trained with the regularization (the second term in Eq. (7)) to learn useful representations during training. In contrast, the decoder $f_{\text{dec}}^{trj}$ of the predictor $f_1$ trained on a trajectory-based dataset is relatively free from the bias and captures a robust mapping from the hidden representation space to ionic transport properties $y_s$. Accordingly, we initialize the decoder $f_{\text{dec}}^{str}$ using $f_{\text{dec}}^{trj}$.

**Training of the predictor $f_2$.** After the initialization, we train the predictor $f_2$ using a structure-based dataset $\mathcal{D}^{str}$. Specifically, we perform optimization by minimizing the $L_1$ loss between the prediction $\hat{y}^j$, computed from the structure–temperature embedding $\mathbf{X}^j$, and the ground truth ionic transport property $y_s^j$, where $j$ denotes the $j$-th data sample in the structure-based dataset $\mathcal{D}^{str}$.

## 4. Experimental Results and Analysis

### 4.1. Experimental Setup

We evaluate our method on three datasets with varying availability of atomic trajectories, data generation methods, ion species, target ionic transport properties, and evaluation temperatures (see Table 2).

**Dataset 1 (trajectory-based dataset $\mathcal{D}^{trj}$).** Dataset 1 includes atomic trajectories obtained from MD simulations (Nam et al., 2025). The target property is the MSD of Li ions, evaluated at the final simulation time step under four different temperatures.

**Dataset 2 (structure-based dataset $\mathcal{D}^{str}$).** We deliberately exclude the atomic trajectories available in the original dataset (Zheng et al., 2024). This setting enables a rigorous evaluation under scenarios where atomic trajectories are unavailable, which reflects the fact that most real-world ionic transport data are reported without trajectory information. Unlike Datasets 1 and 3, which only contain Li ions, Dataset 2 provides ionic diffusivities $\text{D}_s$ for 62 ion species $s$. Among them, Na ions are excluded from the training set and used exclusively for testing to evaluate generalization to an unseen ion species.

**Dataset 3 (real-world structure-based dataset $\mathcal{D}^{str}$).** While Datasets 1 and 2 are generated from MD simulations, Dataset 3 consists of *experimentally* measured ionic conductivity $\sigma_{\text{Li}}$ (Therrien et al., 2025). As direct experimental

*Table 3.* Comparison with benchmark methods on Dataset 1. Performance is evaluated in terms of inference time (s) for a total of 419 materials across four temperatures, and the MAE of the target $\log_{10} \mathrm{MSD_{Li}}$ at each temperature. The best performer is highlighted as **bold**. Statistical significance is evaluated using paired two-tailed t-tests with a significance level of 0.05. Standard deviations are available in Appendix F.1. Additional comparison with compositional and structural feature-based random forest baseline (Therrien et al., 2025) is available in Appendix F.3.

| Type | Methodology | Inference time (s) 419 materials $\times 4T$ | MAE($\log_{10} \mathrm{MSD_{Li}}$) 600K | 800K | 1000K | 1200K |
|---|---|---|---|---|---|---|
| Autoregressive | LiFlow (Nam et al., 2025) | 2910 | 0.378 | 0.392 | 0.457 | 0.407 |
| Non-autoregressive | MatFormer (Yan et al., 2022) | 22 | 0.604 | 0.685 | 0.894 | 1.207 |
| | ComFormer (Yan et al., 2024) | **14** | 0.451 | 0.531 | 0.642 | 0.760 |
| | DenseGNN (Du et al., 2024) | 29 | 0.412 | 0.472 | 0.531 | 0.523 |
| | Ours | **14** | **0.344** | **0.367** | **0.402** | **0.390** |

observation of atomic trajectories is extremely challenging, Dataset 3 is a structure-based dataset $\mathcal{D}^{str}$ and enables evaluation on real-world experimental data.

**Implementation.** All experiments are conducted on a machine with an Intel(R) Core(TM) i9-10920X CPU @ 3.50 GHz and an NVIDIA RTX A6000 GPU. We report the mean absolute error (MAE) as the performance metric, where lower values indicate better performance. Implementation details, including model architectures and optimization, are provided in Appendix E.

### 4.2. Performance on a Trajectory-Based Dataset (Dataset 1)

Table 3 demonstrates that our non-autoregressive learning framework with model-level auxiliary modality learning outperforms both autoregressive (Nam et al., 2025) and non-autoregressive benchmarks (Yan et al., 2022; 2024; Du et al., 2024) in terms of inference time and prediction error on the trajectory-based dataset (*i.e.*, Dataset 1).

**Inference time.** Compared to the autoregressive benchmark (Nam et al., 2025), all non-autoregressive methods (Yan et al., 2022; 2024; Du et al., 2024), including ours, achieve dramatically reduced inference time by directly predicting ionic transport properties instead of performing sequential rollouts of atomic trajectories. Even among non-autoregressive approaches, our model is one of the fastest, despite incorporating large foundation models (Park et al., 2024; Goswami et al., 2024). This efficiency arises from three design choices. First, MOMENT (Goswami et al., 2024), a foundation model for time-series analysis, is not used at inference time. Second, we employ only the early layers of SevenNet (Park et al., 2024), significantly reducing computational cost. Third, the predictor $f_1$ consists of a lightweight linear encoder and an MLP decoder.

**Prediction error.** Non-autoregressive benchmarks (Yan et al., 2022; 2024; Du et al., 2024) typically exhibit higher prediction error than that of the autoregressive model (Nam et al., 2025), as atomic trajectories containing dynamical information are not utilized as input. In contrast, our model-level auxiliary modality learning strategy effectively distills long-time dynamical information from atomic trajectories

into the predictor $f_1$ during training, resulting in lower MAE than even that of the autoregressive baseline (Nam et al., 2025).

**Prediction stability.** We further observe that diffusion-based autoregressive methods (Fu et al., 2023; Schreiner et al., 2023) frequently suffer from divergence in atomic trajectories, where predicted atomic positions explode to unbounded values due to error accumulation, although those methods are not included in Table 3. In contrast, non-autoregressive learning benchmarks, including our approach, avoid trajectory rollouts and therefore provide stable and reliable predictions of ionic transport properties.

### 4.3. Performance on Structure-Based Datasets (Datasets 2–3)

Table 4 shows that our framework with auxiliary modality learning significantly reduces the MAE compared to existing non-autoregressive benchmarks (Yan et al., 2022; 2024; Du et al., 2024) on structure-based datasets that lack atomic trajectories. It is worth noting that autoregressive benchmarks, such as LiFlow (Nam et al., 2025), cannot be trained on structure-based datasets because they inherently require atomic trajectories during training. In contrast, existing non-autoregressive benchmarks exhibit limited performance, as they are unable to exploit dynamic information contained in a trajectory-based dataset. Our framework overcomes this limitation by leveraging dynamic information from a trajectory-based dataset for training on structure-based datasets. Specifically, results on Dataset 2 indicate that our framework generalizes better to the unseen ion species $\mathrm{Na}$, which is not included in the training set. Furthermore, results on Dataset 3 demonstrate that our framework achieves substantially lower prediction error on real-world experimental data compared to existing non-autoregressive benchmarks.

**Practical significance of the results on Dataset 3.** The error on the experimental dataset (Dataset 3) is larger than those observed on the simulation datasets (Datasets 1 and 2), as shown in Tables 3 and 4. However, an error of 1.388 on the $\log_{10}$ scale is comparable to the variability of experimental measurements of ionic conductivity, which typically

*Table 4.* Comparison with benchmark methods on Datasets 2 and 3. For each dataset, performance is evaluated in terms of the MAE of the target definitions, $\log_{10} D_{Na}$ and $\log_{10} \sigma_{Li}$, respectively, at each corresponding temperature. All models are pretrained on Dataset 1 and are subsequently fine-tuned on Datasets 2 or 3. Note that autoregressive models cannot be trained on Datasets 2 and 3 due to the absence of atomic trajectory data. The best performer is highlighted as **bold**. Statistical significance is evaluated using paired two-tailed $t$-tests with a significance level of 0.05. Standard deviations are available in Appendix F.2. Additional comparison with compositional and structural feature-based random forest baseline (Therrien et al., 2025) is available in Appendix F.3.

| METHODOLOGY | DATASET 2: MAE($\log_{10} D_{Na}$) | | | | DATASET 3: MAE($\log_{10} \sigma_{Li}$) |
| --- | --- | --- | --- | --- | --- |
| | 1000K | 1500K | 2000K | 2500K | 300K |
| MATFORMER (YAN ET AL., 2022) | 0.527 | 0.361 | 0.463 | 0.651 | 2.090 |
| COMFORMER (YAN ET AL., 2024) | 0.447 | 0.386 | 0.418 | 0.517 | 2.150 |
| DENSEGNN (DU ET AL., 2024) | 0.616 | 0.491 | 0.368 | 0.312 | 2.048 |
| OURS | **0.166** | **0.149** | **0.074** | **0.064** | **1.388** |

*Table 5.* Ablation study on Dataset 1. The best performer is highlighted as **bold**.

| METHODOLOGY | DATASET 1: MAE($\log_{10} MSD_{Li}$) | | | |
| --- | --- | --- | --- | --- |
| | 600K | 800K | 1000K | 1200K |
| OURS | **0.344** | **0.367** | **0.402** | **0.390** |
| W/O MODEL-LEVEL AUXILIARY MODALITY LEARNING | 0.395 | 0.428 | 0.473 | 0.467 |
| CLOSED-FORM INITIALIZATION → GRADIENT-BASED OPTIMIZATION | 0.354 | 0.406 | 0.455 | 0.421 |
| W/O REGULARIZATION OF $g$ | 0.348 | 0.374 | 0.408 | 0.408 |
| FOURIER + MOMENT → FOURIER | 0.429 | 0.433 | 0.466 | 0.419 |
| SEVENNET → LIFLOW STRUCTURE EMBEDDING | 0.387 | 0.388 | 0.458 | 0.411 |
| W/O POLYNOMIAL EXPANSION | 0.372 | 0.409 | 0.456 | 0.425 |

ranges from approximately 0.5 to 2 orders of magnitude across different studies (Wachter-Welzl et al., 2018; Shimoda et al., 2022). Since our method is intended for initial screening, where the goal is to filter candidate materials prior to more expensive molecular dynamics simulations, this level of error remains practically meaningful.

### 4.4. Ablation Study on a Trajectory-Based Dataset

**Model-level auxiliary modality learning.** The first two rows of Table 5 demonstrate that model-level auxiliary modality learning contributes substantially to performance improvements on a trajectory-based dataset. In the "w/o model-level auxiliary modality learning" setting, the predictor is trained from random initialization without knowledge transfer from the dual-modal trainer $g$. Model-level auxiliary modality learning enables the non-autoregressive predictor $f_1$ to benefit from atomic trajectories through knowledge transfer from the dual-modal trainer $g$, although the predictor $f_1$ itself does not directly consume atomic trajectories as input.

**Closed-form initialization of predictor $f_1$.** The key idea behind model-level auxiliary modality learning is to enforce the predictor $f_1$'s hidden representations to match those of the dual-modal trainer $g$, despite the absence of atomic trajectories as input. This approach is conceptually similar to conventional knowledge distillation (Aguilar et al., 2020), which aligns teacher and student representations via gradient-based optimization by utilizing a loss function quantifying dissimilarity between them. In contrast, our method differs methodologically in that it employs a closed-form initialization that directly matches hidden representations through a direct linear algebraic solver. The

third row of Table 5 indicates that closed-form initialization achieves more effective knowledge transfer than gradient-based optimization, resulting in lower MAE. We attribute this improvement to data scarcity. In this regime, direct solvers can obtain accurate solutions while avoiding optimization error, at an affordable computational cost.

**Regularization of dual-modal trainer $g$.** There is a risk that $\mathbf{W}_{\mathbf{x},T}$ fails to learn informative representations because the dual-modal trainer $g$ may rely overly on the trajectory encoder $\mathbf{W}_{\mathbf{p}}$ due to the stronger supervision signal provided by atomic trajectories. To mitigate this risk, we introduce the regularization term in Eq. (7), which allows the structure encoder $\mathbf{W}_{\mathbf{x},T}$ to contribute meaningfully to prediction. The results in the fourth row of Table 5 confirm that this regularization consistently improves performance on the trajectory-based dataset.

**Analysis of input embeddings.** Our framework further benefits from foundation models, including MOMENT and SevenNet. To assess the contribution of MOMENT, we evaluate a variant in which trajectory embeddings are replaced by Fourier-transformed atomic trajectories without MOMENT. This variant exhibits higher MAE, indicating that the time-series foundation model effectively extracts trajectory embeddings enriched with dynamic information. Similarly, to evaluate the contribution of SevenNet, we replace SevenNet-based structure embeddings with structure embeddings from LiFlow (Nam et al., 2025), the autoregressive benchmark. This replacement leads to increased prediction error, demonstrating that SevenNet provides more informative structure representations from equilibrium atomic structures. Together, these results highlight that our framework successfully leverages the strong generalization capabilities of foundation models, despite the target material

*Table 6.* Ablation study on Datasets 2 and 3. The best performer is highlighted as **bold**. Further ablation study of embedding methods on Datasets 2 and 3 is provided in Appendix F.4.

| METHODOLOGY | DATASET 2: MAE($\log_{10} D_{Na}$) | | | | DATASET 3: MAE($\log_{10} \sigma_{Li}$) |
| --- | --- | --- | --- | --- | --- |
| | 1000K | 1500K | 2000K | 2500K | 300K |
| OURS | **0.166** | 0.149 | **0.074** | **0.064** | **1.388** |
| W/O MODEL-LEVEL AUXILIARY MODALITY LEARNING | 0.201 | **0.146** | 0.089 | 0.078 | 1.412 |
| W/O TWO-LEVEL AUXILIARY MODALITY LEARNING | 0.351 | 0.244 | 0.135 | 0.109 | 1.539 |
| INITIALIZATION WITH $\mathbf{W}_{\mathbf{x},T} \rightarrow \mathbf{W}^{trj}$ | 0.365 | 0.219 | 0.101 | 0.111 | 1.430 |

property prediction task differing from their original training objectives. Finally, removing polynomial expansion from both structure and temperature embeddings in Eqs. (3) and (4), respectively, degrades performance, supporting our hypothesis that polynomial expansion indeed enables modeling nonlinear relationships between input embeddings and hidden representations, even when a linear encoder is used.

### 4.5. Ablation Study on Structure-Based Datasets

**Auxiliary modality learning.** The first three rows of Table 6 demonstrate that both model- and data-level auxiliary modality learning schemes generally lead to lower MAE across temperatures and datasets. Model-level auxiliary modality learning enables the predictor to incorporate atomic dynamics learned from the trajectory-based dataset, while data-level auxiliary modality learning allows this dynamical knowledge to be transferred to structure-based datasets that lack atomic trajectories. Interestingly, although Dataset 1 contains ionic transport properties only for the Li ion species, training on Dataset 1 still improves performance on Dataset 2, which includes multiple ion species. More importantly, Dataset 3 also benefits from Dataset 1, despite substantial differences in both data source (experiments *vs.* MD simulations) and target definition (ionic conductivity *vs.* MSD at the final simulation step). These results indicate that knowledge learned from the trajectory-based dataset is transferable across ion species, data sources, and target formulations.

**Initialization of predictor** $f_2$**.** As illustrated in Figure 2, in data-level auxiliary modality learning, we initialize the encoder $\mathbf{W}^{str}$ using the structure encoder $\mathbf{W}_{\mathbf{x},T}$ of the dual-modal trainer $g$, rather than the encoder $\mathbf{W}^{trj}$ of the predictor $f_1$. This design choice is motivated by the observation that the encoder $\mathbf{W}^{trj}$ may become biased toward the trajectory-based dataset during model-level auxiliary modality learning, as it is enforced to imitate the hidden representations of the dual-modal trainer $g$. As shown in the fourth row of Table 6, initializing with $\mathbf{W}^{trj}$ indeed results in higher prediction error, confirming that naïvely reusing the trajectory-adapted encoder hinders generalization to structure-based datasets.

**Additional experimental results.** We provide additional experimental results on generalization to polymer materials, comparison with joint training, and hyperparameter sensitivity analysis in Appendices F.5, F.6, and F.7, respectively.

## 5. Limitations

Although our framework demonstrates the effectiveness of leveraging scientific foundation models for data-efficient ionic transport prediction, a deeper understanding of how these foundation models affect the proposed framework remains necessary. First, additional experiments in Appendix F.5 show that our method, built upon SevenNet (Park et al., 2024) trained primarily on inorganic materials, can generalize to polymers, a materially distinct class. This suggests the potential generalizability of pretrained scientific foundation models, but a more systematic analysis is needed to clarify when and why such transferability holds. Second, our framework relies on linear encoders to align trajectory embeddings and structure–temperature embeddings with the hidden representation space. We hypothesize that this linearity is closely related to the representation spaces induced by foundation models, similar to observations in vision–language foundation models (Moayeri et al., 2023; Merullo et al., 2023). However, understanding the conditions under which this linear assumption holds, when it breaks down, and how it depends on the choice of foundation models remains an important direction for future work.

## 6. Conclusions and Outlook

We proposed a non-autoregressive learning framework for predicting ionic transport properties that enables fast, accurate, and stable inference without error accumulation by leveraging auxiliary modality learning. Model-level auxiliary modality learning allows trajectory-free inference while utilizing atomic trajectories during training. Our framework further leverages data-level auxiliary modality learning that enables knowledge transfer from a trajectory-based dataset to the training of predictors on structure-based datasets where atomic trajectories are unavailable. We demonstrate that the proposed framework achieves substantial improvements in both inference speed and prediction accuracy across diverse datasets. While this work focuses on ionic transport properties, the proposed framework is readily extensible to other material properties governed by atomic dynamics, thereby offering a general pathway to accelerate MD-based material property prediction and facilitate high-throughput material screening. More broadly, our closed-form initialization can be applied to tasks where data-efficient auxiliary modality learning is necessary.

## Acknowledgments

The work of Y.-W. Shin was supported in part by the National Research Foundation of Korea (NRF), South Korea grant funded by the Korea government (MSIT) (RS-2021-NR059723) and by SMEs Technology Innovation Development Program through the Technology Innovation and Promotion Agency (TIPA), funded by Ministry of SMEs and Startups (RS-2024-00511332). The work of B. Lee was supported by Institutional Projects at the Korea Institute of Science and Technology (26E0223).

## Impact Statement

This paper presents work whose goal is to advance the field of machine learning for scientific modeling, with a particular focus on enabling fast and accurate prediction of ionic transport properties from equilibrium atomic structures. The proposed non-autoregressive learning framework built upon auxiliary modality learning could help reduce the computational cost and energy footprint of large-scale materials screening, thereby accelerating the discovery of ion-conducting materials relevant to rechargeable batteries and related energy technologies. Beyond battery applications, the framework is broadly applicable to other properties governed by atomic dynamics and may reduce the cost of MD-based property estimation in a range of materials science and engineering settings. We note that model predictions can be unreliable under distribution shift; thus, high-stakes decisions should validate top candidates via independent simulations or experiments.

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

# A. Related Work

## A.1. Autoregressive MD Acceleration

MD simulations rely on numerical integration to advance atomic configurations in time, which typically requires extremely small integration time steps (*e.g.*, $\sim 10^{-15}$ s) to ensure numerical stability (Zheng et al., 2021). Such short time steps result in a large number of simulation steps and consequently high computational cost. Autoregressive MD acceleration frameworks (Hsu et al., 2024; Schreiner et al., 2023; Klein et al., 2023; Nam et al., 2025; Li et al., 2024; Yu et al., 2024) aim to overcome this limitation by avoiding explicit numerical integration and instead learning to predict future atomic configurations directly using generative models. These approaches enable the use of effective time steps that exceed the stability limits of classical integrators. Representative generative paradigms for this purpose include normalizing flows (Klein et al., 2023), diffusion models (Hsu et al., 2024; Schreiner et al., 2023), flow matching (Nam et al., 2025; Li et al., 2024), and bridge matching (Yu et al., 2024). Most existing approaches have primarily focused on biological applications, modeling the dynamical trajectories of biomolecules such as peptides and proteins. These studies typically target conformational dynamics in relatively small systems in solution, rather than crystalline solids, where the scientific objective is often to extract macroscopic transport coefficients from long-time trajectories. In solid-state materials, transport commonly arises from rare activated hopping events (Rodin et al., 2022; Gustafsson et al., 2024), which pose distinct challenges compared to biomolecular folding dynamics. To the best of our knowledge, LiFlow (Nam et al., 2025) is the first work explicitly developed from a materials perspective to target ionic transport properties in solid-state systems. It substantially reduces the computational cost of MD through a flow-matching-based predictor and further mitigates error accumulation using a flow-matching-based corrector. Nevertheless, because ionic transport properties are obtained by first generating atomic trajectories in an autoregressive manner and subsequently estimating transport coefficients, the overall procedure remains computationally demanding, and accumulated errors may still lead to inaccurate or even unstable long-time trajectories.

## A.2. Non-Autoregressive Material Property Prediction

Material property prediction has been extensively studied, resulting in a broad spectrum of machine learning models (Xie & Grossman, 2018; Schütt et al., 2017; Chen et al., 2019; Choudhary & DeCost, 2021; Du et al., 2024; Yan et al., 2022; 2024). These models are built upon convolutional architectures (Xie & Grossman, 2018; Schütt et al., 2017), graph neural networks (Chen et al., 2019; Choudhary & DeCost, 2021; Du et al., 2024), and transformer-based backbones (Yan et al., 2022; 2024). However, their inputs typically consist solely of static atomic configurations, which prevents them from explicitly accounting for atomic dynamics. For instance, the widely used MatBench (Dunn et al., 2020) benchmark targets the prediction of material properties such as phonon-related quantities, elastic moduli, formation energies, band gaps, and metallicity from equilibrium atomic structures. These properties are either static or depend only weakly on long-time or long-range atomic motion. In comparison, ionic transport is governed by long-time correlated motion and rare diffusion events, which are commonly quantified through time-correlation functions or mean-squared displacement accumulated over extended trajectories; such information cannot be inferred from a single equilibrium configuration alone. Another popular benchmark dataset, MD17 (Chmiela et al., 2017), focuses on predicting energies and forces from snapshots of atomic configurations. Although MD17 contains atomic trajectories, the associated learning objectives are restricted to instantaneous quantities and do not require modeling long-time dynamics or emergent transport behavior. Consequently, despite substantial progress in materials property prediction, learning-based approaches that explicitly leverage atomic dynamics to infer long-time transport properties in solid-state systems remain scarce.

## A.3. Illustrative Comparison of Ionic Transport Prediction Methods

Comparison of existing method and our framework for ionic transport prediction is summarized in Figure 3.

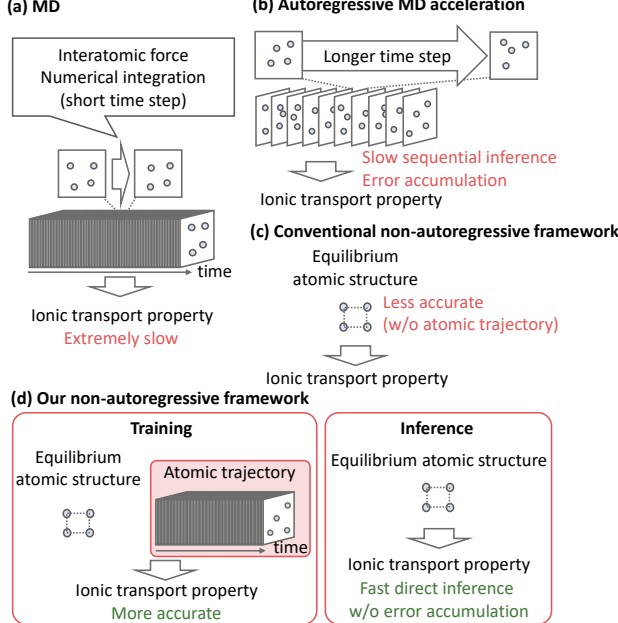

*Figure 3.* Comparison of existing methods and our framework for ionic transport prediction. **(a) MD simulation** explicitly evolves atomic trajectories by numerically integrating interatomic forces using extremely small time steps, requiring a large number of simulation steps to obtain transport-relevant statistics. **(b) Autoregressive MD acceleration** increases the time step size but still relies on sequential trajectory rollouts, resulting in slow inference and error accumulation. **(c) Conventional non-autoregressive learning frameworks** directly predict material properties from equilibrium structures, enabling fast inference but often sacrificing accuracy due to the absence of dynamical information. **(d) Our non-autoregressive learning framework** exploits atomic trajectories only during training to achieve higher accuracy, while allowing fast and stable inference without atomic trajectories.

# B. Notations

Table 7 summarizes the notations used throughout the paper.

*Table 7.* Summary of notations

| Notation | Description |
| --- | --- |
| $\mathbf{x}$ | Equilibrium atomic structure |
| $\mathbf{p}$ | Atomic trajectory |
| $T$ | Temperature |
| $y_s$ | Ionic transport property for ionic species $s$ |
| $\mathrm{MSD}_s$ | Mean squared displacement of ionic species $s$ at the final simulation time |
| $\mathrm{D}_s$ | Ionic diffusivity of species $s$ |
| $\sigma_s$ | Ionic conductivity of species $s$ |
| $\mathcal{D}^{trj}$ | Trajectory-based dataset |
| $\mathcal{D}^{str}$ | Structure-based dataset |
| $\mathbf{E}_{\mathbf{p}}^i$ | Trajectory embedding of the $i$-th sample |
| $\mathbf{E}_{\mathbf{x}}^i$ | Structure embedding of the $i$-th sample |
| $\mathbf{E}_T^i$ | Temperature embedding of the $i$-th sample |
| $\mathbf{X}^i$ | Structure–temperature embedding of the $i$-th sample |
| $f_1, f_2$ | Predictor |
| $g$ | Dual-modal trainer |
| $\mathbf{W}_{\mathbf{p}}$ | Trajectory encoder of the dual-modal trainer |
| $\mathbf{W}_{\mathbf{x},T}$ | Structure encoder of the dual-modal trainer |
| $\mathbf{W}^{trj}$ | Encoder of the predictor trained on the trajectory-based dataset |
| $\mathbf{W}^{str}$ | Encoder of the predictor adapted for the structure-based dataset |
| $\mathbf{H}_{\mathbf{p}}^i$ | Trajectory-based hidden representation of the $i$-th sample |
| $\mathbf{H}_{\mathbf{x},T}^i$ | Structure-based hidden representation of the $i$-th sample |
| $\mathbf{H}^i$ | Combined hidden representation of the $i$-th sample |
| $g_{dec}$ | Decoder of the dual-modal trainer |
| $f_{dec}^{trj}$ | Decoder of the predictor trained on the trajectory-based dataset |
| $f_{dec}^{str}$ | Decoder of the predictor adapted for the structure-based dataset |
| $\lambda_b$ | Balancing parameter for regularization of the dual-modal trainer |
| $\lambda_r$ | Ridge regularization parameter |
| $\mathcal{L}(\cdot, \cdot)$ | $L_1$ loss function |
| $\odot$ | Element-wise multiplication |
| $[\cdot\,;\cdot]$ | Concatenation |
| $\mathcal{N}^i(k)$ | Neighbor set of atom $k$ in the $i$-th sample |
| $\mathbf{E}_a^i$ | Atom embedding of the $i$-th sample |
| $\mathbf{n}_k^i$ | Node embedding of the $k$-th atom for the $i$-th sample |
| $\mathbf{m}_{kl}^i$ | Edge embedding corresponding to the interaction between atoms $k$ and $l$ for the $i$-th sample |

## C. Pseudocodes for Our Non-Autoregressive Learning Framework

We first present the pseudocode for our model-level auxiliary modality learning in our non-autoregressive learning framework.

---

**Algorithm 1** Model-Level Auxiliary Modality Learning

---

**Require:** Trajectory-based dataset $\mathcal{D}^{trj} = \{(\mathbf{p}^i, \mathbf{x}^i, T^i, y_s^i)\}_{i=1}^{|\mathcal{D}^{trj}|}$
**Require:** Dual-modal trainer $g = \{\mathbf{W_p}, \mathbf{W}_{\mathbf{x},T}, g_{dec}\}$
**Require:** Predictor $f_1 = \{\mathbf{W}^{trj}, f_{dec}^{trj}\}$

1: **Stage 0. Pre-compute embeddings**
2: **for** each sample $(\mathbf{p}^i, \mathbf{x}^i, T^i, y_s^i) \in \mathcal{D}^{trj}$ **do**
3:     $\mathbf{E}_{\mathbf{p}}^i \leftarrow \text{MOMENT}(\mathbf{p}^i)$ (Goswami et al., 2024)
4:     $\mathbf{E}_{\mathbf{x}}^i \leftarrow \text{SEVENNET}(\mathbf{x}^i)$ (Park et al., 2024)
5: **end for**

6: **Stage 1: Train the dual-modal trainer $g$**
7: **for** each epoch **do**
8:     **for** each $(\mathbf{E}_{\mathbf{p}}^i, \mathbf{E}_{\mathbf{x}}^i, T^i, y_s^i) \in \mathcal{D}^{trj}$ **do**
9:         $\mathbf{E}_T^i \leftarrow \text{TEMPEMBED}(T^i)$
10:         $\mathbf{X}^i \leftarrow [\mathbf{E}_{\mathbf{x}}^i; \mathbf{E}_T^i]$
11:         $\mathbf{H}_{\mathbf{p}}^i \leftarrow \mathbf{E}_{\mathbf{p}}^i \mathbf{W_p}$
12:         $\mathbf{H}_{\mathbf{x},T}^i \leftarrow \mathbf{X}^i \mathbf{W}_{\mathbf{x},T}$
13:         $\mathbf{H}^i \leftarrow \mathbf{H}_{\mathbf{p}}^i + \mathbf{H}_{\mathbf{x},T}^i$
14:         $\hat{y}^i \leftarrow g_{dec}(\mathbf{H}^i)$
15:         $\hat{y}_{\mathbf{x},T}^i \leftarrow g_{dec}(\mathbf{H}_{\mathbf{x},T}^i)$
16:         Update $g = \{\mathbf{W_p}, \mathbf{W}_{\mathbf{x},T}, g_{dec}\}$ by minimizing $\mathcal{L}(\hat{y}^i, y_s^i) + \lambda_b \mathcal{L}(\hat{y}_{\mathbf{x},T}^i, y_s^i)$
17:     **end for**
18: **end for**

19: **Stage 2: Closed-form initialization of the predictor $f_1$**
20: Initialize encoder $\mathbf{W}^{trj} \leftarrow \left( \sum_{i=1}^{|\mathcal{D}^{trj}|} (\mathbf{X}^i)^\top \mathbf{X}^i + \lambda_r \mathbf{I} \right)^{-1} \left( \sum_{i=1}^{|\mathcal{D}^{trj}|} (\mathbf{X}^i)^\top \mathbf{H}^i \right)$.
21: Initialize decoder $f_{dec}^{trj} \leftarrow g_{dec}$
22: **Stage 3: Fine-tune predictor $f_1$**
23: **for** each epoch **do**
24:     **for** each $(\mathbf{E}_{\mathbf{p}}^i, \mathbf{E}_{\mathbf{x}}^i, T^i, y_s^i) \in \mathcal{D}^{trj}$ **do**
25:         $\mathbf{E}_T^i \leftarrow \text{TEMPEMBED}(T^i)$
26:         $\mathbf{X}^i \leftarrow [\mathbf{E}_{\mathbf{x}}^i; \mathbf{E}_T^i]$
27:         $\hat{y}^i \leftarrow f_{dec}^{trj}(\mathbf{X}^i \mathbf{W}^{trj})$
28:         Update $f_1 = \{\mathbf{W}^{trj}, f_{dec}^{trj}\}$ by minimizing $\mathcal{L}(\hat{y}^i, y_s^i)$
29:     **end for**
30: **end for**
31: **Return trained predictor $f_1$**

---

We also present the pseudocode for data-level auxiliary modality learning in our non-autoregressive learning framework.

---

**Algorithm 2** Data-Level Auxiliary Modality Learning

---

**Require:** Structure-based dataset $\mathcal{D}^{str} = \{(\mathbf{x}^j, T^j, y_s^j)\}_{j=1}^{|\mathcal{D}^{str}|}$
**Require:** Structure encoder $\mathbf{W}_{\mathbf{x},T}$ of the dual-modal trainer trained on $\mathcal{D}^{trj}$
**Require:** Decoder $f_{dec}^{trj}$ of the predictor trained on $\mathcal{D}^{trj}$
 1: **Stage 0. Pre-compute embeddings**
 2: **for** each sample $(\mathbf{x}^j, T^j, y_s^j) \in \mathcal{D}^{str}$ **do**
 3:     $\mathbf{E}_{\mathbf{x}}^j \leftarrow$ SEVENNET$(\mathbf{x}^j)$ (Park et al., 2024)
 4: **end for**
 5: **Stage 1: Initialization**
 6: $\mathbf{W}^{str} \leftarrow \mathbf{W}_{\mathbf{x},T}$
 7: $f_{dec}^{str} \leftarrow f_{dec}^{trj}$
 8: **Stage 2: Training of predictor** $f_2$ **on** $\mathcal{D}^{str}$
 9: **for** each epoch **do**
10:     **for** each $(\mathbf{E}_{\mathbf{x}}^j, T^j, y_s^j) \in \mathcal{D}^{str}$ **do**
11:         $\mathbf{E}_T^j \leftarrow$ TEMPEMBED$(T^j)$
12:         $\mathbf{X}^j \leftarrow [\mathbf{E}_{\mathbf{x}}^j; \mathbf{E}_T^j]$
13:         $\hat{y}^j \leftarrow f_{dec}^{str}(\mathbf{X}^j \mathbf{W}^{str})$
14:         Update $f_2 = \{\mathbf{W}^{str}, f_{dec}^{str}\}$ by minimizing $\mathcal{L}(\hat{y}^j, y_s^j)$
15:     **end for**
16: **end for**
17: **Return trained predictor** $f_2$

---

## D. Ionic Transport Properties

Depending on the datasets, target ionic transport property $y_s$ may correspond to the mean squared displacement $\mathrm{MSD}_s$ (Nam et al., 2025), the ionic diffusivity $\mathrm{D}_s$ (Zheng et al., 2024), or the ionic conductivity $\sigma_s$ (Therrien et al., 2025). These quantities describe different aspects of the same underlying ion transport process.

**MSD.** For an ionic species $s$, the mean squared displacement is defined as the mean of the squared displacement traveled by ionic species $s$ over time $t$:

$$\mathrm{MSD}_s(t) = \left\langle \|\mathbf{p}_s(t) - \mathbf{p}_s(0)\|^2 \right\rangle, \tag{10}$$

where $\mathbf{p}_s(t)$ denotes the position of ion $s$ at time $t$ and $< \cdot >$ indicates averaging over ionic species $s$.

**Ionic Diffusivity.** Ionic diffusivity quantifies the intrinsic mobility of ions through thermally driven random motion. In the long-time diffusive regime, the ionic diffusivity $\mathrm{D}_s$ is obtained from the slope of the MSD via the Einstein relation:

$$\mathrm{D}_s = \lim_{t \to \infty} \frac{1}{6} \frac{d}{dt} \mathrm{MSD}_s(t). \tag{11}$$

**Ionic Conductivity.** Ionic conductivity measures the macroscopic charge transport induced by an external electric field. The ionic diffusivity $\mathrm{D}_s$ is related to the ionic conductivity $\sigma_s$ through the Nernst–Einstein relation:

$$\sigma_s = \frac{n_s q_s^2}{k_B T} \mathrm{D}_s, \tag{12}$$

where $n_s$ is the number density of mobile ions of species $s$, $q_s$ is the ionic charge, $k_B$ is the Boltzmann constant, and $T$ is the temperature. The Nernst–Einstein relation assumes uncorrelated ion motion. In practice, ion–ion correlations and collective effects may lead to deviations.

# E. Implementation Details

In this section, we present a detailed technical description of the proposed non-autoregressive learning framework built upon auxiliary modality learning in Section 3.

## E.1. Foundation Models

We extract node and edge embeddings from the 10th layer of SevenNet-0 (Park et al., 2024), which provides structural representations at a relatively low computational cost compared to other MLIPs. These embeddings are then aggregated to form the structure embedding. For the trajectory embedding, we employ Moment-1-small (Goswami et al., 2024), a foundation model designed for time-series data, which provides a dedicated interface for obtaining trajectory-level embeddings.

## E.2. Training of Dual-Modal Trainer $g$

**Architecture of the dual-modal trainer $g$.** The dual-modal trainer $g$ consists of two linear encoders and a shared nonlinear decoder. The trajectory encoder $\mathbf{W_p}$ maps the trajectory embedding $\mathbf{E_p}^i \in \mathbb{R}^{d_\mathbf{p}}$ to a hidden representation of dimension $d_h$, while the structure encoder $\mathbf{W}_{\mathbf{x},T}$ maps the structure–temperature embedding $\mathbf{X}^i \in \mathbb{R}^{d_{\mathbf{x},T}}$ to the same latent space:

$$\mathbf{W_p} \in \mathbb{R}^{d_\mathbf{p} \times d_h}, \quad \mathbf{W}_{\mathbf{x},T} \in \mathbb{R}^{d_{\mathbf{x},T} \times d_\mathbf{h}}.$$

Here, $d_\mathbf{p}$, $d_{\mathbf{x},T}$, and $d_h$ denote the dimensions of the trajectory embedding, the structure–temperature embedding, and the hidden representation, respectively. In all experiments, we set $d_h = 8$.

The two hidden representations are combined additively and normalized using a LayerNorm without affine parameters. The decoder $g_{dec}$ is implemented as a four-layer multilayer perceptron (MLP) with ReLU activations. Specifically, the decoder consists of three hidden layers of width 4,000 followed by a linear output layer that predicts a scalar ionic transport property. The same decoder is used to produce predictions from the combined latent representation and from the structure-based hidden representation for regularization.

**Optimization.** The dual-modal trainer $g$ is optimized using the Adam optimizer with a learning rate of $10^{-3}$ for 50 epochs. The training objective consists of the sum of two $L_1$ loss terms in Eq. (7) corresponding to the combined prediction and the structure-based prediction. The balancing parameter $\lambda_b$ between the two loss terms is set to 1.0.

## E.3. Closed-Form Initialization

**Architecture of the predictor $f$.** The predictor $f_1$ consists of a linear encoder $\mathbf{W}^{trj} \in \mathbb{R}^{d_x T \times d_h}$ followed by LayerNorm and an MLP decoder $f_{dec}^{trj}$. The decoder architecture is identical to that of the dual-modal trainer and is initialized using the pretrained decoder $g_{dec}$.

**Ridge regularization.** The ridge regularization parameter in the closed-form initialization (see Eq. (9)) is set to $\lambda_r = 10^{-5}$ for all experiments.

## E.4. Fine-Tuning of Predictor $f_1$

After closed-form initialization, the predictor $f_1$ is fine-tuned on the trajectory-based dataset using Adam with a learning rate of $10^{-5}$ for 50 epochs.

## E.5. Data-Level Auxiliary Modality Learning

**Dataset 2.** During training on Dataset 2, the encoder's initial learning rate is set to $10^{-2}$, while the initial learning rate of the first two layers of the decoder is set to $10^{-4}$. The initial learning rate of the remaining layers in the decoder is set to $10^{-6}$. Training is performed using the Adam optimizer for 100 epochs. For each epoch, the learning rate is decreased by 1%.

**Dataset 3.** During training on Dataset 3, the encoder's initial learning rate is set to $10^{-2}$, while the initial learning rate of the first two layers of the decoder is set to $10^{-4}$. The initial learning rate of the remaining layers in the decoder is set to $10^{-6}$. Training is performed using the Adam optimizer for 1000 epochs. For each epoch, the learning rate is decreased by 0.1%.

# F. Additional Experimental Results

The following table summarizes the contents of this section.

*Table 8.* Summary of additional experimental results

| Contents |
| --- |
| F.1. Performance on a trajectory-based dataset (Dataset 1) |
| F.2. Performance on structure-based datasets (Datasets 2 and 3) |
| F.3. Comparison with a random forest baseline |
| F.4. Ablation study on Datasets 2 and 3 |
| F.5. Generalization to polymer materials |
| F.6. Comparison with joint training |
| F.7. Hyperparameter sensitivity analysis |

### F.1. Performance on a Trajectory-Based Dataset (Dataset 1)

Table 9 extends Table 3 by additionally reporting the standard deviation alongside the mean.

*Table 9.* Comparison with benchmark methods on Dataset 1. Performance is evaluated in terms of inference time (s) for a total of 419 materials across four temperatures, and the MAE of the target $\log_{10} \mathrm{MSD_{Li}}$ at each temperature. The best performer is highlighted as **bold**. Statistical significance is evaluated using paired two-tailed t-tests with a significance level of 0.05.

| METHODOLOGY | INFERENCE TIME (S) 419 MATERIALS $\times 4T$ | MAE($\log_{10} \mathrm{MSD_{Li}}$) 600K | 800K | 1000K | 1200K |
|---|---|---|---|---|---|
| LiFlow | $2910 \pm 59$ | $0.378 \pm 0.006$ | $0.392 \pm 0.007$ | $0.457 \pm 0.007$ | $0.407 \pm 0.010$ |
| MatFormer | $22 \pm 0.2$ | $0.604 \pm 0.020$ | $0.685 \pm 0.008$ | $0.894 \pm 0.013$ | $1.207 \pm 0.024$ |
| ComFormer | $\mathbf{14 \pm 0.6}$ | $0.451 \pm 0.131$ | $0.531 \pm 0.136$ | $0.642 \pm 0.235$ | $0.760 \pm 0.418$ |
| DenseGNN | $29 \pm 1.6$ | $0.412 \pm 0.006$ | $0.472 \pm 0.006$ | $0.531 \pm 0.024$ | $0.523 \pm 0.017$ |
| Ours | $\mathbf{14 \pm 0.5}$ | $\mathbf{0.344 \pm 0.006}$ | $\mathbf{0.367 \pm 0.004}$ | $\mathbf{0.402 \pm 0.005}$ | $\mathbf{0.390 \pm 0.005}$ |

### F.2. Performance on Structure-Based Datasets (Datasets 2 and 3)

Table 10 extends Table 4 by additionally reporting the standard deviation alongside the mean.

*Table 10.* Comparison with benchmark methods on Datasets 2 and 3. For each dataset, performance is evaluated in terms of the MAE of the target definitions, $\log_{10} \mathrm{D_{Na}}$ and $\log_{10} \sigma_{\mathrm{Li}}$, respectively, at each corresponding temperature. All models are pretrained on Dataset 1 and are subsequently fine-tuned on Datasets 2 or 3. Note that autoregressive models cannot be trained on Datasets 2 and 3 due to the absence of atomic trajectory data. The best performer is highlighted as **bold**. Statistical significance is evaluated using paired two-tailed $t$-tests with a significance level of 0.05.

| METHODOLOGY | DATASET 2: MAE($\log_{10} \mathrm{D_{Na}}$) 1000K | 1500K | 2000K | 2500K | DATASET 3: MAE($\log_{10} \sigma_{\mathrm{Li}}$) 300K |
|---|---|---|---|---|---|
| MatFormer | $0.527 \pm 0.000$ | $0.361 \pm 0.000$ | $0.463 \pm 0.001$ | $0.651 \pm 0.001$ | $2.090 \pm 0.000$ |
| ComFormer | $0.447 \pm 0.080$ | $0.386 \pm 0.042$ | $0.418 \pm 0.114$ | $0.517 \pm 0.221$ | $2.150 \pm 0.204$ |
| DenseGNN | $0.616 \pm 0.141$ | $0.491 \pm 0.108$ | $0.368 \pm 0.103$ | $0.312 \pm 0.100$ | $2.048 \pm 0.181$ |
| Ours | $\mathbf{0.166 \pm 0.018}$ | $\mathbf{0.149 \pm 0.027}$ | $\mathbf{0.074 \pm 0.012}$ | $\mathbf{0.064 \pm 0.019}$ | $\mathbf{1.388 \pm 0.078}$ |

### F.3. Comparison with a Random Forest Baseline

Tables 11 and 12 compare our method with a random forest model (Therrien et al., 2025) based on compositional and structural features on Datasets 1–3. Our method consistently outperforms the baseline across all temperatures and datasets, highlighting its advantage in capturing transport-relevant information beyond static descriptors.

*Table 11.* Comparison with a random forest baseline on Dataset 1. The best performer is highlighted as **bold**.

| METHODOLOGY | DATASET 1: MAE($\log_{10} \text{MSD}_{\text{Li}}$) | | | |
| | 600K | 800K | 1000K | 1200K |
| --- | --- | --- | --- | --- |
| RANDOM FOREST | $0.465 \pm 0.005$ | $0.582 \pm 0.005$ | $0.705 \pm 0.006$ | $0.816 \pm 0.007$ |
| OURS | $\mathbf{0.344 \pm 0.006}$ | $\mathbf{0.367 \pm 0.004}$ | $\mathbf{0.402 \pm 0.005}$ | $\mathbf{0.390 \pm 0.005}$ |

*Table 12.* Comparison with a random forest baseline on Datasets 2 and 3. The best performer is highlighted as **bold**.

| METHODOLOGY | DATASET 2: MAE($\log_{10} \text{D}_{\text{Na}}$) | | | | DATASET 3: MAE($\log_{10} \sigma_{\text{Li}}$) |
| | 1000K | 1500K | 2000K | 2500K | 300K |
| --- | --- | --- | --- | --- | --- |
| RANDOM FOREST | $0.368 \pm 0.029$ | $0.354 \pm 0.015$ | $0.284 \pm 0.020$ | $0.227 \pm 0.025$ | $1.608 \pm 0.078$ |
| OURS | $\mathbf{0.166 \pm 0.018}$ | $\mathbf{0.149 \pm 0.027}$ | $\mathbf{0.074 \pm 0.012}$ | $\mathbf{0.064 \pm 0.019}$ | $\mathbf{1.388 \pm 0.078}$ |

## F.4. Ablation Study on Datasets 2 and 3

Table 13 extends Table 6 by additionally including two cases: (1) replacing the SevenNet (Park et al., 2024) embedding with the LiFlow (Nam et al., 2025) embedding, and (2) omitting the polynomial expansion of embeddings in Eqs. (3) and (4). The results demonstrate that both foundation model–based structure embeddings and polynomial expansion contribute to performance gains across datasets.

In contrast to Table 5, we do not include ablations on closed-form initialization, regularization of $g$, and MOMENT (Goswami et al., 2024), as Datasets 2 and 3 are structure-based datasets without trajectory information, making these components inapplicable.

*Table 13.* Ablation study on Datasets 2 and 3. The best performer is highlighted as **bold**.

| METHODOLOGY | DATASET 2: MAE($\log_{10} D_{Na}$) | | | | DATASET 3: MAE($\log_{10} \sigma_{Li}$) |
|---|---|---|---|---|---|
| | 1000K | 1500K | 2000K | 2500K | 300K |
| OURS | **0.166** | 0.149 | **0.074** | **0.064** | **1.388** |
| W/O MODEL-LEVEL AUXILIARY MODALITY LEARNING | 0.201 | **0.146** | 0.089 | 0.078 | 1.412 |
| W/O TWO-LEVEL AUXILIARY MODALITY LEARNING | 0.351 | 0.244 | 0.135 | 0.109 | 1.539 |
| INITIALIZATION WITH $\mathbf{W}_{\mathbf{x},T} \to \mathbf{W}^{trj}$ | 0.365 | 0.219 | 0.101 | 0.111 | 1.430 |
| SEVENNET $\to$ LIFLOW STRUCTURE EMBEDDING | 0.416 | 0.490 | 0.561 | 0.477 | 1.963 |
| W/O POLYNOMIAL EXPANSION | 0.274 | 0.164 | 0.150 | 0.132 | 1.621 |

### F.5. Generalization to Polymer Materials

We further evaluate performance of our auxiliary modality learning framework on a polymer dataset (Xie et al., 2023; Xie et al.), which represents a distinct material class compared to the inorganic systems (Datasets 1–3). We use a subset of 355 materials selected from a total of 5,962 materials by sorting them according to their sample IDs and taking the first 355 entries.

First, we treat the polymer dataset as a trajectory-based dataset to examine whether model-level auxiliary modality learning remains effective in this setting. The results in Table 14 demonstrate that our method outperforms the variant without auxiliary modality learning, demonstrating that model-level auxiliary modality learning generalizes beyond inorganic systems. However, for polymers, preprocessing atomic trajectories with a Fourier transform prior to obtaining trajectory embeddings using MOMENT (Goswami et al., 2024) leads to degraded performance. This may be attributed to fundamental differences in atomic motion between crystalline materials (Dataset 1) and amorphous polymer materials.

Second, we treat the polymer dataset as a structure-based dataset to evaluate whether the data-level auxiliary modality learning strategy generalizes to polymer systems. The results indicate that data-level auxiliary modality learning improves accuracy even in this setting.

These behaviors are likely due to the strong generalization capability of the underlying foundation model (Ju et al., 2025). A theoretical understanding of this capability remains an important direction for future work.

*Table 14.* Performance on polymer materials.

| METHODOLOGY | MAE($\log_{10} D_{Li}$) |
|---|---|
| OURS (AS TRAJECTORY-BASED DATASET) W/O FOURIER TRANSFORMATION | 0.131 |
| OURS (AS TRAJECTORY-BASED DATASET) | 0.142 |
| OURS (AS STRUCTURE-BASED DATASET) | 0.170 |
| W/O AUXILIARY MODALITY LEARNING | 0.210 |

## F.6. Comparison with Joint Training

We compare our method with a jointly (*e.g.,* simultaneously) trained model on Datasets 1 and 2. The results in Table 15 show that our auxiliary modality learning approach consistently outperforms the case of joint training.

Specifically, data-level auxiliary modality learning is advantageous over joint training from three perspectives:

• **Missing trajectory information.** Dataset 2 lacks atomic trajectories, limiting the use of dynamic information in joint training.

• **Dataset inhomogeneity.** Differences in simulation conditions introduce inconsistencies that are not explicitly handled in joint training.

• **Target mismatch.** Joint training requires conversion between target properties (*e.g.,* ionic conductivity to diffusivity), which can cause additional error.

*Table 15.* Comparison with joint training. The best performer is highlighted as **bold**.

| METHODOLOGY | DATASET 1: MAE($\log_{10} \mathrm{MSD_{Li}}$) | | | | DATASET 2: MAE($\log_{10} \mathrm{D_{Na}}$) | | | |
| --- | --- | --- | --- | --- | --- | --- | --- | --- |
| | 600K | 800K | 1000K | 1200K | 1000K | 1500K | 2000K | 2500K |
| JOINT TRAINING | 0.376 | 0.414 | 0.445 | 0.449 | 0.560 | 0.198 | 0.142 | 0.150 |
| OURS | **0.344** | **0.367** | **0.402** | **0.390** | **0.166** | **0.149** | **0.074** | **0.064** |

## F.7. Hyperparameter Sensitivity Analysis

As shown in Figure 4 (a), the performance of our framework is robust to the choice of the ridge regularization parameter $\lambda_r$, with less than 10% variation in MAE, while achieving consistent gains from auxiliary modality learning across a wide range (1e-3 to 1e-7) on Dataset 1. Figures 4 (b) and (c) further show consistent gains from auxiliary modality learning, indicating that our framework is not sensitive to $\lambda_r$ across datasets. Based on this observation, we select $\lambda_r$ based on Dataset 1, where the closed-form initialization is directly applied, rather than tuning it separately for each structure-based dataset.

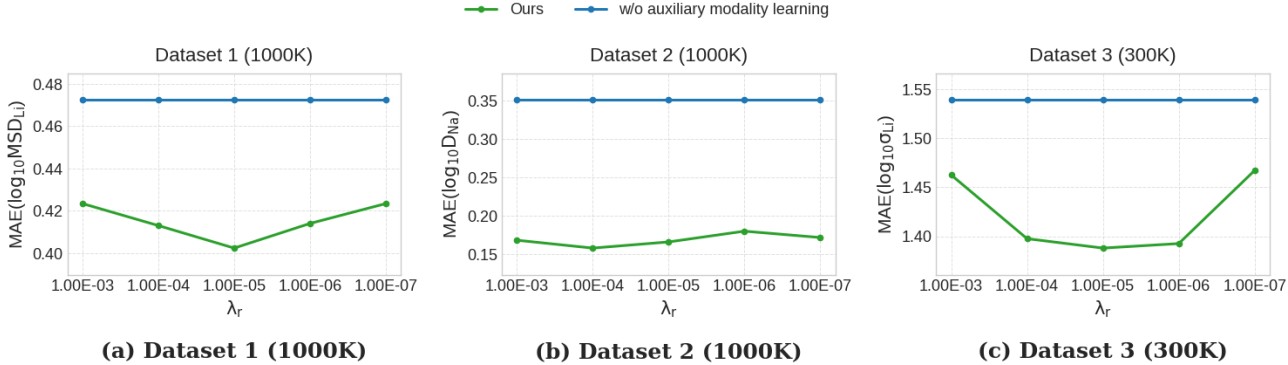

**(a) Dataset 1 (1000K)**      **(b) Dataset 2 (1000K)**      **(c) Dataset 3 (300K)**

*Figure 4.* Sensitivity analysis of the ridge regularization parameter $\lambda_r$. (a) Dataset 1 (1000K) (b) Dataset 2 (1000K) (c) Dataset 3 (300K)

