# OpenReview forum: "Teaching Molecular Dynamics to a Non-Autoregressive Ionic Transport Predictor"
_ICML.cc/2026/Conference — ICML 2026 regular_

### Official Review · Reviewer_okya · 2026-03-11

**Soundness:** 3
**Presentation:** 3
**Significance:** 3
**Originality:** 3
**Overall Recommendation:** 3
**Confidence:** 3

**Summary:**

This paper proposes a non-autoregressive learning setup by doing modality reduction so that ionic transport properties can be predicted just using a static atomic structure by taking the trajectories of those atoms into account but discarding them once done as an auxiliary modality; the transfer happens by closed-form ridge regression initialization taking dynamical knowledge out of a dual-modal trainer into the low-weight predictor, then optionally extends it for structure-only data with trajectory-less information with some dataset-modalities, which results in more than a 200× speedup relative to autoregressive baselines and reduced errors across different benchmark datasets.

**Compliance With Llm Reviewing Policy:**

Affirmed.

**Key Questions For Authors:**

1. The closed-form initialization relies on ridge regression to match hidden representations. How sensitive is final prediction performance to the choice of \lambda_r and was any systematic tuning performed across datasets?
2. The framework is evaluated only on lithium and sodium ion systems. Does the modality reduction approach generalize to materials with significantly different ionic transport mechanisms?

**Limitations:**

The ablation study is only conducted on Dataset 1. It is unclear whether the same design choices remain optimal for structure-based datasets with different ionic species or target properties.

**Strengths And Weaknesses:**

Soundness: The technical claims are well supported by ablation studies and comparisons across multiple datasets, and the closed-form ridge regression initialization is mathematically grounded; however, the evaluation is limited to relatively small datasets.
Presentation: The paper is clearly structured with pseudocode, notation tables, and illustrative figures that make the framework easy to follow.
Significance: Predicting ionic transport properties efficiently is practically important for battery materials discovery, and the proposed framework meaningfully reduces both computational cost and prediction error.
Originality: The core idea of treating trajectories as a training-only auxiliary modality and transferring knowledge via closed-form initialization is a novel combination that is clearly distinguished from conventional knowledge distillation and standard non-autoregressive prediction.

---

> ### Author Rebuttal · Authors · 2026-03-31
>
> Dear Reviewer Okya,
>
> Thank you for your constructive feedback.
>
> # W1. Small dataset size
> The relatively small dataset size arises from the high cost of MD simulations and experimental measurements, which makes large-scale data collection impractical. Our modality reduction framework is specifically designed to address this data scarcity by effectively leveraging limited trajectory data and transferring its information to structure-based prediction.
>
> # Q1. Sensitivity to $\lambda_r$
> As shown in Fig.M1-4 in our link [https://anonymous.4open.science/r/author2](https://anonymous.4open.science/r/author2), the performance is robust to the choice of $\lambda_r$, with less than 10% variation in MAE and consistent gains from modality reduction across a wide range(1e-3 to 1e-7) on Dataset 1. Fig.M5-6 further show consistent gains, indicating that the framework is not sensitive to $\lambda_r$ across datasets.
>
> Based on this observation, we select $\lambda_r$ using Dataset 1, where the closed-form initialization is directly applied, rather than tuning it separately for each structure-based dataset.
>
>
>
> # Q2. Generalization to polymers
> We further evaluate our modality reduction framework on a polymer dataset [1], which represents a distinct material class compared to the inorganic systems (Datasets 1–3). Due to time constraints, we use a subset of 355 materials; although limited in size, it is sufficient to observe consistent trends.
>
> We conduct two complementary experiments. First, we treat the polymer dataset as a trajectory-based dataset to evaluate model-modality reduction. Second, we treat it as a structure-based dataset to evaluate dataset-modality reduction when transferring from Dataset 1 to a different material class.
>
> In both settings, our method consistently outperforms the variant without modality reduction, demonstrating that the proposed framework generalizes beyond the specific ionic systems considered in the main experiments.
>
>
> |Method|MAE|
> |--|--|
> |Ours(as trajectory-based dataset)|0.142|
> |Ours(as structure-based dataset)|0.170|
> |w/o modality reduction|0.210|
>
> These results suggest that our modality reduction mechanism learns general structure-dynamics relationship rather than relying on system-specific ionic transport mechanisms.
>
> [1] T. Xie et al. A cloud platform for sharing and automated analysis of raw data from high throughput polymer MD simulations, APL Machine Learning (2023)
>
> # Correction of Results on Dataset 2
> We identified an issue in the data split for Dataset 2 (Tables 4 and 6), where the training and test sets were not properly separated. This has been corrected, and all experiments have been rerun using the proper split. We further verified statistical significance via t-tests. Importantly, despite this correction, the overall trends and relative performance compared to baselines remain consistent, and our conclusions are unchanged.
>
> ## Table 4
> |Method|1000K|1500K|2000K|2500K|
> |--|--|--|--|--|
> |Matformer|0.527|0.361|0.463|0.651|
> |Comformer|0.447|0.386|0.418|0.517|
> |DenseGNN|0.616|0.491|0.368|0.312|
> |Ours|0.166|0.149|0.074|0.064|
>
> ## Table 6
> |Method|1000K|1500K|2000K|2500K|
> |--|--|--|--|--|
> |Ours|0.166|0.149|0.074|0.064|
> |w/o model modality reduction|0.201|0.146|0.089|0.078|
> |w/o both modality reduction|0.351|0.244|0.135|0.109|
> |Initialization with W_trj|0.365|0.219|0.101|0.111|
>
> # L1. Ablation study on Datasets 2 and 3
> Ablation results on Datasets 2 and 3 are originally reported in Table 6 of our paper and also presented below. The results show that each component—modality reduction and the proposed initialization scheme—contributes to the overall performance.
>
> We further evaluate variants using SevenNet-based structure embeddings replaced with LiFlow embeddings and without polynomial expansion. The results consistently demonstrate that both foundation model–based structure embeddings and polynomial expansion contribute to performance gains across datasets.
>
> We do not include ablations on closed-form initialization, regularization of g, and MOMENT, as Datasets 2 and 3 are structure-based datasets without trajectory information, making these components inapplicable.
>
> |Method|1000K|1500K|2000K|2500K|Dataset3(300K)|
> |--|--|--|--|--|--|
> |Ours|0.166|0.149|0.074|0.064|1.388|
> |w/o model modality reduction|0.201|0.146|0.089|0.078|1.412|
> |w/o both modality reduction|0.351|0.244|0.135|0.109|1.539|
> |Initialization with W_trj|0.365|0.219|0.101|0.111|1.430|
> |SevenNet->Liflow|0.416|0.490|0.561|0.477|1.963|
> |w/o polynomial expansion|0.274|0.164|0.150|0.132|1.621|

---

> > ### Author Rebuttal · Reviewer_okya · 2026-04-03
> >
> > I would like to thank all the authors for solving my problems and addressing my concerns. I will adjust my rating based on the actual situation.

---

> > > ### Author Response · Authors · 2026-04-05
> > >
> > > Dear Reviewer okya,
> > >
> > > We sincerely thank you for your careful review and thoughtful feedback. We are glad that our responses helped address your concerns and clarify the key aspects of our work. We truly appreciate your re-evaluation of the paper. Thank you again for your time and support.

---

### Official Review · Reviewer_Ryyw · 2026-03-12

**Soundness:** 3
**Presentation:** 3
**Significance:** 3
**Originality:** 3
**Overall Recommendation:** 4
**Confidence:** 2

**Summary:**

The paper proposes a non-autoregressive framework that employs model-modality reduction to capture atomic dynamics during training while enabling trajectory-free inference. During training, a dual-modal trainer uses both structure and trajectory embeddings, then a predictor is applied via closed-form ridge regression to mimic the trainer's hidden representations using only structure-temperature features. Experiments demonstrate substantial improvements in both prediction accu
racy and inferencespeed over strong non-autoregressive baselines.

**Compliance With Llm Reviewing Policy:**

Affirmed.

**Final Justification:**

My concerns have been addressed.

**Key Questions For Authors:**

- Have you considered training a unified predictor on multiple datasets jointly and comparing its performance against your proposed approach? The experiment would help further justify the necessity of the staged modality reduction.

- Could you elaborate on the implementation details of the “W/O MODEL-MODALITY REDUCTION” ablation reported in Table 5?

**Limitations:**

yes

**Strengths And Weaknesses:**

- Well motivated problem with clear formulations: The paper explicitly identify the modality gap between static inputs and dynamic targets. By doing so, it effectively highlights why prior approaches struggle to simultaneously achieve low computational cost and high accuracy.
- Innovative methodology: By aligning hidden representations via closed-form ridge regression, the model avoids gradient optimization errors on small datasets and improves feature transfer accuracy.
- Strong generalization: The proposed framework significantly outperforms baseline methods in prediction accuracy on real-world experimental data and unseen ion species.
- Dependency on pretrained models: Despite the lightweight design of the predictor , its predictive performance relies heavily on two pretrained foundation models.

---

> ### Author Rebuttal · Authors · 2026-03-31
>
> Dear Reviewer Ryyw,
>
> Thank you for your positive feedback and for recognizing the strengths of our work.
>
> # Correction of Results on Dataset 2
> We identified an error in the data split for Dataset 2 (Tables 4 and 6), where the train and test sets were not properly separated, and have corrected it. All experiments have been rerun with the proper split, and statistical significance has been verified via t-tests. The updated results show consistent trends and do not affect the conclusions of the paper.
>
> ## Table 4
> |Method|1000K|1500K|2000K|2500K|
> |--|--|--|--|--|
> |Matformer|0.527|0.361|0.463|0.651|
> |Comformer|0.447|0.386|0.418|0.517|
> |DenseGNN|0.616|0.491|0.368|0.312|
> |Ours|0.166|0.149|0.074|0.064|
>
> ## Table 6
> |Method|1000K|1500K|2000K|2500K|
> |--|--|--|--|--|
> |Ours|0.166|0.149|0.074|0.064|
> |w/o model modality reduction|0.201|0.146|0.089|0.078|
> |w/o both modality reduction|0.351|0.244|0.135|0.109|
> |Initialization with W_trj|0.365|0.219|0.101|0.111|
> # W1. Dependency on pretrained models
> While our method relies on pretrained foundation models, this is a deliberate design choice. By decoupling representation learning from the predictor, our framework can leverage rich physical priors encoded in foundation models while keeping the downstream predictor lightweight and data-efficient. This design also allows our method to directly benefit from future improvements in foundation models without retraining the entire pipeline.
>
> # Q1. Comparison with joint training
> We compare our method with a jointly trained model on Datasets 1 and 2. The results show that our modality-reduction approach consistently outperforms simultaneous joint training.
> |Method|600K(D1)|800K(D1)|1000K(D1)|1200K(D1)|1000K(D2)|1500K(D2)|2000K(D2)|2400K(D2)|
> |--|--|--|--|--|--|--|--|--|
> |joint training|0.376|0.414|0.445|0.449|0.560|0.198|0.142|0.150|
> |ours|0.344|0.367|0.402|0.390|0.166|0.149|0.074|0.064|
>
> Data-Modality reduction is advantages over joint training in three perspective:
> - Missing trajectory information: Dataset 2 lacks atomic trajectories, limiting the use of dynamic information in joint training.
> - Dataset inhomogeneity: Differences in simulation conditions introduce inconsistencies that are not explicitly handled in joint training.
> - Target mismatch: Joint training requires conversion between target properties (e.g., ionic conductivity to diffusivity), which can introduce additional error.
>
>
> # Q2. W/o model-modality reduction
> In the “w/o model-modality reduction” setting, the predictor is trained from random initialization without knowledge transfer from the dual-modal trainer. As a result, it does not benefit from trajectory-informed representations, leading to degraded performance compared to our full model.

---

> > ### Author Rebuttal · Reviewer_Ryyw · 2026-04-02
> >
> > I thank the authors for their thoughtful rebuttal and appreciate the clarifications provided. While the response is helpful, it does not fundamentally change my overall assessment, and I will therefore maintain my original score.

---

> > > ### Author Response · Authors · 2026-04-05
> > >
> > > Dear Reviewer Ryyw,
> > >
> > > We sincerely thank you for your positive assessment and constructive feedback. We are glad that our responses helped clarify the key aspects of our work and address your concerns. Thank you again for your time and support.

---

### Official Review · Reviewer_gTDV · 2026-03-13

**Soundness:** 3
**Presentation:** 3
**Significance:** 3
**Originality:** 3
**Overall Recommendation:** 4
**Confidence:** 3

**Summary:**

This paper proposes a non-autoregressive learning framework for predicting ionic transport properties (MSD, diffusivity, ionic conductivity) from static equilibrium atomic structures. The key idea is "modality reduction": treating atomic trajectories as an auxiliary modality available only during training. A dual-modal trainer is first trained on both trajectory and structure embeddings (using MOMENT and SevenNet foundation models, respectively). Knowledge is then transferred to a trajectory-free predictor via closed-form ridge regression initialization that aligns hidden representations, followed by fine-tuning. An optional dataset-modality reduction step transfers knowledge from trajectory-based to structure-based datasets. Experiments on three datasets (one trajectory-based, two structure-based including experimental data) show improvements over autoregressive (LiFlow) and non-autoregressive (Matformer, Comformer, DenseGNN) baselines in both accuracy and inference speed (200x over LiFlow).

**Compliance With Llm Reviewing Policy:**

Affirmed.

**Final Justification:**

Raised my scores to 3s and weak accept after reviews

**Key Questions For Authors:**

**1. How does this framework relate to the LUPI (Learning Using Privileged Information) paradigm?**
Could the authors discuss how modality reduction differs from or extends Vapnik & Vashist (2009), generalized distillation (Lopez-Paz et al., 2016), and auxiliary modality learning (Shen et al., ICML 2023)?
*Impact:* Properly contextualizing the contribution would significantly affect the originality assessment.

**2. What are the standard deviations across multiple runs?**
How many random seeds were used? What is the variance of results, especially on the small Dataset 1?
*Impact:* Would determine whether improvements over baselines are statistically robust beyond the claimed t-tests.

**3. How does the method compare to simple physics-based ionic transport predictors?**
For example, bond valence site energy methods or random forest on compositional features (as benchmarked in the OBELiX paper)?
*Impact:* Would clarify the practical advantage of the deep learning framework over simpler domain approaches.

**4. What error level is acceptable for practical materials screening on Dataset 3?**
Is an MAE of ~1.4 on log10 sigma sufficient to rank candidate electrolytes? How does this compare to the typical spread of experimental measurements across different labs?
*Impact:* Would help assess practical significance.

**5. When does the linear encoder assumption break down?**
Are there material classes or transport regimes where the linear projection from structure embeddings to trajectory-informed representations fails?
*Impact:* Would clarify the scope and limitations of the approach.

**Limitations:**

Partially addressed. The paper mentions unreliability under distribution shift (Impact Statement, Section 5), which is appropriate. However, the discussion does not address: (i) the large absolute error on experimental data, (ii) the reliance on specific foundation models (SevenNet, MOMENT) that may not generalize to all material systems, (iii) the linearity assumption's potential failure modes, or (iv) the assumption that trajectory-based knowledge transfers across ion species and data sources (which is demonstrated empirically but not analyzed theoretically).

**Strengths And Weaknesses:**

### 2. Strengths

**1. Well-motivated problem with practical relevance.**
Ionic transport prediction is a genuine bottleneck in battery materials screening. The paper clearly articulates the modality gap between static inputs and dynamic targets, and why both autoregressive (slow, error accumulation) and naive non-autoregressive (no dynamics) approaches fall short. This is an underexplored area in ML for materials.

**2. Principled dual-modality reduction framework.**
The two-stage design (model-modality reduction for trajectory-based data, dataset-modality reduction for structure-based data) is well-engineered. The additive combination of trajectory and structure representations (Eq. 6) with the balancing regularization (Eq. 7) that prevents the trainer from ignoring structure embeddings is a thoughtful design choice, validated by ablation (Table 5, row 4).

**3. Closed-form initialization is elegant for the data-scarce regime.**
Using ridge regression (Eq. 9) rather than gradient-based distillation is well-suited to the small-data setting of ionic transport. The ablation in Table 5 (row 3) confirms this provides better knowledge transfer than iterative optimization, and the justification via direct solver accuracy in the small-N regime is convincing.

**4. Effective use of foundation models.**
Leveraging SevenNet (an MLIP) for physics-informed structure embeddings and MOMENT for time-series trajectory embeddings is a creative design choice that achieves strong performance with a lightweight linear encoder on top.

**5. Comprehensive evaluation across diverse datasets.**
Testing on three datasets spanning MD-simulated and experimentally measured targets, multiple ion species (Li, Na, 62 others), different transport quantities (MSD, diffusivity, conductivity), and temperatures from 300K to 2500K provides a thorough evaluation. The generalization to unseen Na ions in Dataset 2 and real experimental data in Dataset 3 is particularly noteworthy.

**6. Thorough ablation studies.**
Tables 5 and 6 systematically ablate each component: modality reduction, closed-form vs. gradient-based initialization, regularization, foundation model contributions, and polynomial expansion. This clearly demonstrates what drives performance.

---

### 3. Weaknesses

**1. Missing connection to Learning Using Privileged Information (LUPI) literature.**
The core idea of "modality reduction" -- using auxiliary information at training time but not at inference -- is the well-established LUPI framework (Vapnik & Vashist, 2009; Lopez-Paz et al., 2016). The closely related "Auxiliary Modality Learning" (Shen et al., ICML 2023) is also uncited. The paper presents modality reduction as a novel concept without positioning against this substantial body of work. The brief comparison to knowledge distillation in Section 4.4 is insufficient -- it discusses only representation alignment but misses that the broader paradigm of training with privileged modalities and distilling to unimodal students is well-explored.

**2. No variance reporting across runs.**
Despite mentioning paired two-tailed t-tests for statistical significance, the paper reports no standard deviations or confidence intervals in Tables 3-6. It is unclear how many random seeds were used, or what the variance of results is. This is particularly important given the small dataset sizes (Dataset 1 has only 419 materials split across train/test), where results may be sensitive to random initialization.

**3. No domain-standard baselines beyond ML methods.**
All baselines are ML architectures (Matformer, Comformer, DenseGNN, LiFlow). Missing are: (i) simple physics-informed descriptors for ionic conductivity (e.g., bond valence site energy methods or activation energy from nudged elastic band calculations), (ii) simple compositional/structural feature-based regression (e.g., random forest on hand-crafted descriptors as in the OBELiX benchmark paper), and (iii) direct MLIP-MD simulations which are the practical alternative. Without these, it is difficult to assess the practical value of the ML approach vs. established domain workflows.

**4. Large prediction error on experimental Dataset 3.**
The MAE of 1.388 on log10 sigma_Li (Table 4) means predictions can deviate by more than an order of magnitude from experimental values. While this is better than baselines (~2.0), it raises questions about practical utility for materials screening. The paper would benefit from discussing what error level is acceptable for the intended application (e.g., ranking materials for experimental validation) and whether this error is comparable to experimental uncertainties or variance among DFT/MD predictions.

**5. Strong linearity assumptions are not justified from a physical perspective.**
The encoder is restricted to a single linear layer (Eq. 5), with nonlinearity introduced only through polynomial expansion of embeddings (Eq. 3-4). The implicit assumption that the trajectory's contribution to ionic transport can be linearly projected from structure-temperature embeddings (Eq. 8-9) is a strong physical claim that is not discussed or justified. While the ablation shows this works empirically, understanding when this linear approximation breaks down would strengthen the contribution.

**6. Terminology may confuse readers.**
"Modality reduction" suggests dimensionality reduction or compression, when the actual operation is modality dropout/ablation at inference. More standard terminology (e.g., "privileged modality learning," "training-time auxiliary modality") would improve clarity and help readers connect to the existing literature.

---

> ### Author Rebuttal · Authors · 2026-03-31
>
> Dear Reviewer gTDV,
>
> We greatly appreciate your insightful and constructive feedback.
> # W1, Q1 & W6: Related work, terminology, and novelty
> ## Related work and terminology
> Thank you for pointing out the relevant literature. We agree that our framework can be viewed as an extension of the LUPI paradigm, where atomic trajectories serve as privileged information available only during training. LUPI (Vapnik & Vashist, 2009) was later unified with knowledge distillation (Hinton et al., 2015) into generalized distillation (Lopez-Paz et al., 2016), and further extended to auxiliary modality learning (Shen et al., ICML 2023). We will revise the introduction and related work to explicitly position our method within this line of research and adopt the appropriate terminology (auxiliary modality learning).
> ## Novelty
> Our work has extended the LUPI paradigm in three aspects:
> - Application to materials domains: To the best of our knowledge, this is the first work that formulates atomic trajectories as a privileged modality for ionic transport prediction, where trajectory data is typically unavailable at inference time.
> - Closed-form initialization: We have introduced ridge regression–based closed-form initialization, which is particularly effective in data-scarce regimes, as also reflected in your comment (S3).
> - Alignment across scientific foundation models: To our knowledge, this is the first work that applies the LUPI paradigm at the level of scientific foundation models.
> # W2 & Q2: Variance
> First, the 419 materials correspond only to the test split and Dataset 1 contains 4186 materials in total. Next, we have reported mean and standard deviation over 5 random seeds for Dataset 1 and Dataset 3 (the smallest dataset). Our method exhibits low variance even on Dataset 3.
>
> |Method|Time|600K|800K|1000K|1200K|Dataset 3(300K)|
> |--|--|--|--|--|--|--|
> |Liflow|2910/59|0.378/0.006|0.392/0.007|0.457/0.007|0.407/0.010| |
> |Matformer|22/0.2|0.604/0.020|0.685/0.008|0.894/0.013|1.207/0.024|2.090/0.000|
> |Comformer|14/0.6|0.451/0.131|0.531/0.136|0.642/0.235|0.760/0.418|2.150/0.204|
> |DenseGNN|29/1.6|0.412/0.006|0.472/0.006|0.531/0.024|0.523/0.017|2.048/0.181|
> |Ours|14/0.5|0.344/0.006|0.367/0.004|0.402/0.005|0.390/0.005|1.388/0.078|
> # W3 & Q3: Comparison with random forest (RF)
> Due to limited time and expensive MLIP-MD, we have compared our method with a RF model. Our method consistently outperforms RF across all temperatures and datasets (Dataset 2 omitted due to space), highlighting its advantage in capturing transport-relevant information beyond static descriptors.
> |Method|600K|800K|1000K|1200K|Dataset 1(300K)|
> |--|--|--|--|--|--|
> |RF|0.465/0.005|0.582/0.005|0.705/0.006|0.816/0.007|1.608/0.078|
> |ours|0.344/0.006|0.367/0.004|0.402/0.005|0.390/0.005|1.388/0.078|
> # W4, Q4 & L1: Practical error limit
> It has been reported that experimental measurements of ionic conductivity can vary by approximately 0.5–2 orders of magnitude across different studies [1,2]. In addition, the discrepancy between MD simulations and experimental results is typically on the order of 1 magnitude [3]. In this context, our method is intended for initial screening, where the goal is to preliminarily filter candidate materials before more expensive MD simulations. An error of 1.388 in log10 scale is therefore comparable to the intrinsic uncertainty of the overall simulation–experiment pipeline.
>
> [1] A. Wachter-Welzl et al., solid state ionics (2018)
>
> [2] M. Shimoda et al., Chemistry of Materials (2022)
>
> [3] J. Qi et al., Materials Today Physics (2021)
> # W5, Q5, L2, L3, and L4: Linearity, generalization, and material classes
> - Linearity and generalization: Similar linear alignment phenomena have also been reported across vision–language foundation models [4,5]. Thus, this linearity is highly related to foundation models. We think that this linear approximation may break down when the underlying foundation model(SevenNet0) fails to generalize (due to out of distributions).
> - Generalization across material classes: We have additionally observed that our method, based on SevenNet0 trained on inorganic datasets, generalizes to polymers with significantly different transport mechanisms (see our response to Reviewer okya’s Q2). This behavior is attributed to the strong generalization capability of the underlying foundation model [6], although a theoretical understanding of such cross-domain transfer remains an important direction as future work. We will explicitly state this limitation into the main text.
>
> [4] M. Moayeri et al., ICML (2023)
>
> [5] J. Merullo et al., ICLR (2023)
>
> [6] S. Ju et al., Digital Discovery (2025).
> # Correction of results on Dataset 2
> We identified a code error in the data split for Dataset 2 used in Tables 4 and 6 and have corrected it. All experiments have been re-run with the proper split. The updated results exhibit consistent trends and do not affect the conclusions of the paper (Tables are included in our response to Reviewer Ryyw).

---

> > ### Author Rebuttal · Reviewer_gTDV · 2026-04-04
> >
> > The authors have improved the depth of the research and adequately explained their decision in response to my comments

---

> > > ### Author Response · Authors · 2026-04-05
> > >
> > > Dear Reviewer gTDV,
> > >
> > > We sincerely thank you for your time, careful reading, and insightful feedback. We are glad that our responses helped clarify the key aspects of our work and address your concerns. Your constructive comments have been invaluable in improving both the clarity and rigor of the paper. Thank you again for helping us strengthen the work.

---

### Decision · Program_Chairs · 2026-04-30

**Decision:**

Accept (regular)

**Comment:**

This paper proposes a non-autoregressive framework for predicting the ionic transport properties of materials, aiming to overcome the high computational costs of traditional molecular dynamics simulations and the error accumulation issues found in autoregressive models. The methodology utilizes a modality reduction approach that enables efficient and accurate predictions by leveraging datasets both with and without atomic trajectories. While initial reviews raised questions concerning the model's prediction accuracy and specific design choices, the overall approach was recognized for its technical soundness and innovation in the field of AI-driven materials science.

The authors provided a comprehensive rebuttal that addressed the reviewers' concerns regarding baseline comparisons and the robustness of the modality reduction strategy. A positive consensus was reached among the committee, as the authors' responses effectively clarified the primary technical ambiguities. Although one reviewer was unable to formally update their score or participate in the final discussion due to external constraints, the general sentiment remains favorable towards the paper's contribution.

Therefore, the final decision is a weak accept, with the understanding that the authors will incorporate the additional experimental results and technical clarifications from the rebuttal period into the final version of the manuscript.